# Long-Term Clinical Outcome and Prognostic Factors of Children and Adolescents with Localized Rhabdomyosarcoma Treated on the CWS-2002P Protocol

**DOI:** 10.3390/cancers14040899

**Published:** 2022-02-11

**Authors:** Ewa Koscielniak, Bernd Blank, Christian Vokuhl, Bernarda Kazanowska, Ruth Ladenstein, Felix Niggli, Gustaf Ljungman, Rupert Handgretinger, Guido Seitz, Jörg Fuchs, Birgit Fröhlich, Monika Scheer, Rüdiger Wessalowski, Irene Schmid, Monika Sparber-Sauer, Thomas Klingebiel

**Affiliations:** 1Pediatrics 5 (Oncology, Hematology, Immunology), Klinikum Stuttgart-Olgahospital, 70174 Stuttgart, Germany; b.blank@klinikum-stuttgart.de (B.B.); m.sparber-sauer@klinikum-stuttgart.de (M.S.-S.); 2Faculty of medicine, University of Tübingen, 72016 Tübingen, Germany; 3Section of Pediatric Pathology, Department of Pathology, University Bonn, 53127 Bonn, Germany; christian.vokuhl@ukbonn.de; 4Department of Pediatric Hematology/Oncology and BMT, University of Wroclaw, 50556 Wroclaw, Poland; b.kazanowska@mypost.pl; 5Pediatric Oncology, St. Anna Kinderspital, 1090 Vienna, Austria; ruth.ladenstein@ccri.at; 6Department of Pediatric Oncology, University of Zürich, 8032 Zurich, Switzerland; felix.niggli@uzh.ch; 7Department of Women’s and Children’s Health, Pediatric Oncology, Uppsala University, 75185 Uppsala, Sweden; gustaf.ljungman@kbh.uu.se; 8Department of Pediatric Hematology and Oncology, University of Tübingen, Hospital for Children and Adolescents, 72076 Tuebingen, Germany; rupert.handgretinger@med.uni-tuebingen.de; 9Department of Pediatric Surgery, University Children’s Hospital Marburg, 35043 Marburg, Germany; guido.seitz@med.uni-marburg.de; 10Department of Pediatric Surgery and Urology, Hospital for Children and Adolescents, University Tuebingen, 72076 Tübingen, Germany; joerg.fuchs@med.uni-tuebingen.de; 11Department of Pediatric Hematology and Oncology, University Hospital Münster, 48149 Muenster, Germany; birgit.froehlich@ukmuenster.de; 12Department of Pediatric Oncology and Hematology, Charité-Universitätsmedizin, 13353 Berlin, Germany; monika.scheer@charite.de; 13Department of Pediatric Oncology, Medical Faculty, Heinrich Heine University, Hematologyand Clinical Immunology, 40225 Duesseldorf, Germany; wessalowski@med.uni-duesseldorf.de; 14Department of Pediatrics, Division of Pediatric Hematology and Oncology, Dr. von Hauner Children’s Hospital, University Hospital Munich, LMU Munich, 80337 Munich, Germany; irene.schmid@med.uni-muenchen.de; 15Department for Children and Adolescents, University Hospital Frankfurt, Goethe University, 60590 Frankfurt, Germany; thomas.klingebiel@kinderkrebs-frankfurt.de

**Keywords:** rhabdomyosarcoma, pediatric, soft tissue sarcoma, clinical trial, maintenance therapy, risk grouping

## Abstract

**Simple Summary:**

The major challenge in pediatric oncology is the optimal adaptation of therapy burden to risk profile, aiming to achieve the best outcome with minimum toxicities. The CWS-2002P study in patients ≤ 21 years with localized rhabdomyosarcoma was developed with this goal by reducing or intensifying the chemotherapy depending on the risk group. An important additional aim was to investigate the use of low-dose maintenance chemotherapy. The risk stratification system was effective in predicting outcomes in the four risk groups with very good long-term results. Neither the reduction nor the intensification of chemotherapy influenced the outcome in comparison to previous studies showing that further de-escalation of chemotherapy should be investigated. The weighting of risk factors used for therapy stratification needs to be reevaluated. Maintenance therapy seemed to have an impact on prognosis.

**Abstract:**

We report here the results of the prospective, non-randomized, historically controlled CWS-2002P study in patients ≤ 21 years with localized RMS developed with the aim to improve the long-term outcome by adapting the burden of therapy to risk profile and to investigate the feasibility and relation to the outcome of maintenance therapy (MT) in the high-risk groups. Patients were allocated into low-risk (LR), standard-risk (SR), high-risk (HR), and very high-risk (VHR) groups. Chemotherapy consisted of vincristine (VCR) and dactinomycin (ACTO-D) for all patients with the addition of ifosfamide (IFO) in the SR, HR, and VHR and doxorubicin (DOX) in the HR and VHR groups. Low-dose cyclophosphamide and vinblastine maintenance therapy (MT) over 6 months was recommended in the HR and VHR groups. A total of 444 patients have been included in this analysis. With a median follow-up of 9·6 years (IQR 7·6–10·9) for patients alive, the 5-year EFS and OS for the whole group was 73% (95% CI 69–77) and 80% (95% CI 76–84), respectively. The 5-year EFS by risk group was 100% in the LR, 79% (95% CI 72–84) in the SR, 69% (95% CI 63–75) in the HR, and 42% (95% CI 23–61) in the VHR (log-rank *p* = 0.000). The 5-year EFS was 77% (95% CI 70–84) for 155 patients in the HR group who received MT as compared to 63% (95% CI 50–76) for 49 patients who did not (log-rank *p* = 0.015). Neither the reduction in the IFO dose in the SR nor the increased dose intensity of DOX in HR groups influenced the outcome when compared to the previous CWS and other European studies. MT was feasible, seemed to have an impact on prognosis, and should be studied in a well-controlled prospective trial in this patient population. The weighting of risk factors used for therapy stratification needs to be reevaluated.

## 1. Introduction

Rhabdomyosarcoma (RMS) is the most common soft tissue sarcoma in the first two decades of life. The major progress in survival has been achieved by multicenter international cooperative studies, which showed the efficacy of multimodal therapy [1]. Survival has improved steadily over the last three decades for patients with localized disease and now exceeds 70%. Many outcome-related factors such as histologic subtype, site, age, and size have been defined and included in risk-adapted stratification systems [2,3,4]. In the first CWS studies, the concept of therapy intensification was followed, and all patients with RMS received four drugs cycles: cyclophosphamide (CYC) or IFO, ACTO-D, VCR, and DOX (VACA or VAIA) [5,6,7]. In the following studies, the risk grouping was optimized, adding additional outcome-related factors (Appendix A), and the therapy intensity and the cumulative doses were reduced for the groups with a low and standard risk of recurrence without deterioration of prognosis [8]. The tumor volume reduction after neoadjuvant chemotherapy, which has been shown to be related to prognosis, was included for the secondary risk stratification [6,7].

CWS-2002P was the fifth study of the Cooperative Weichteilsarkomstudie (CWS) Group for children and adolescents with soft tissue sarcoma (STS). It was an overarching study for patients with localized RMS, “RMS-like” soft tissue sarcoma (which included extraskeletal Ewing sarcoma, synovial sarcoma, and undifferentiated sarcoma), and all types of non-rhabdomyosarcoma soft tissue sarcoma (NRSTS). The CWS-2002P concept for the treatment of RMS patients was essentially based on the results of the previous CWS and other European studies, in particular, on the findings from the CWS-96, RMS-96 of the Italian Soft Tissue Sarcoma Committee (STSC), and MMT-95 study, which were conducted as European cooperative projects with complementary objectives [1,7,8,9].

Further risk adaptation of therapy was realized with a reduction in the cumulative IFO dose in the part of the SR group and an increase in the cumulative dose and dose intensity of DOX in the HR and VHR groups in the first 16 weeks of therapy.

## 2. Material and Methods

The CWS-2002P study included prospective, non-randomized, historically controlled trials on localized RMS, “RMS-like” and NRSTS conducted between 17 January 2003 and 31 December 2010 in Austria, Germany, Poland, Sweden, and Switzerland. This analysis included patients with RMS.

The risk grouping based on the common retrospective analysis of European protocols, included additionally to histology, tumor site, IRS group, lymph nodes (LN) involvement also age and tumor size (Table 1), was introduced in the CWS-2002P and in the EpSSG RMS 2005 study [10,11] which were conducted almost contemporarily. The inclusion of tumor size and patient’s age allowed for the selection of patients ≤ 10 years and small tumors (≤5 cm) treated in the previous studies as HR group, who had similar outcomes to the SR group and have been shifted into the SR group in the CWS-2002P.

Patients aged up to 21 years with a diagnosis of localized RMS were eligible provided they had no prior treatment except surgery and started chemotherapy within 8 weeks of the date of biopsy or resection. All diagnoses were confirmed by central pathology review and classified according to the International Rhabdomyosarcoma Classification [12]. Molecular confirmation by FISH (FOXO 1 break) or/and RT-PCR (Foxo1-Pax3/7, Pax3-NCOA1/2) was recommended to diagnose the tumor as alveolar rhabdomyosarcoma (aRMS). If the testing was not performed, the tumor was diagnosed based on morphological criteria. The overall scheme for treatment schedules used and the details of the chemotherapy doses is shown in Figure 1. Patients in the SR group were stratified after four I^2^VA cycles depending on the remission status and primary site in two subgroups, A and B (definitions Figure 1). Subgroup A received subsequently only three VA cycles and subgroup B five I^2^VA cycles. The therapy duration was the same in both arms. In the HR and VHR group, the MT with cyclophosphamide and vinblastine (CYC/VBL) was recommended as an option at the end of the multimodal intensive therapy for patients in complete remission (CR). The dosage of MT should be adapted to the leukocyte count, which should not be below 1500/µL during MT since the continuous administration of MT was the main objective. Chemotherapy dosage was reduced for infants: *<*6 months: 1/3 reduction in the dose calculated by body weight, ≥6 months ≤12 months (or ≤10 kg body weight): 1/3 reduction in the dose calculated per m^2^.

Legend: V vincristine 1.5 mg/m^2^/d; maximum, 2 mg/day, I^2^Ifosfamide 3 g/m^2^/d, on two consecutive days, a dactinomycin 1.5 mg/m^2^/d maximum, 1.5 mg/day, ad doxorubicin 40 mg/m^2^/day, on two consecutive days, MT, maintenance therapy, (optionally for patients who achieved CR at the end of multimodal standard therapy): Cyclophosphamide/Vinblastine: seven 3-weeks cycles of intravenous vinblastine 3 mg/m^2^ on days 1, 8, and 15 and oral cyclophosphamide 2 × 25 mg/m^2^/day days 1–21, with one week pause between the cycles. The dosage of MT should be adapted to the leukocyte count, which should not be below 1500/µL.

Staging procedures included magnetic resonance imaging (MRI) of the primary site and, if indicated, computed tomography (CT) and ultrasound. Metastatic disease was assessed by chest CT, optionally whole-body MRI, cerebral MRI or CT, technetium bone scan, and bone marrow aspiration (2–4 sites). Routine biopsy or lymph node sampling was not required but was recommended for clinically and radiologically suspicious regional lymph nodes. CSF examination was required for all parameningeal head and neck tumors and paravertebral tumors adjacent to the meninges.

The TNM classification was applied to differentiate pre-treatment and postsurgical stages [13]. The clinical staging system (IRS I, II, III) adapted from the Intergroup Rhabdomyosarcoma Study (IRS) postsurgical grouping system was used to categorize patients according to primary surgery [14]. Resection was classified as R0 (free resection margins), R1 (microscopically incomplete), or R2 (macroscopically incomplete).

Tumor volume at diagnosis and after three chemotherapy cycles (7–10 weeks), in patients with measurable disease (IRS group III), was estimated by measuring the maximum diameter of the primary tumor in three dimensions (X, Y, and Z) according to the formula V = 1/6 × π × X × Y × Z and used for tumor response evaluation defined as complete remission (CR complete disappearance of the tumor), good response (GR) ≥ 2/3, poor response (PR) < 2/3 and ≥1/3, objective response (OR) < 1/3 tumor volume reduction progressive disease (PD), any increase in tumor volume, or stable disease.

These response categories were used for stratification of the radiotherapy dose based on the results of previous CWS studies, showing that the tumor volume reduction after neoadjuvant chemotherapy impacts outcome [6,7,15].

A cut-off at 50% tumor volume reduction after three cycles of chemotherapy (7–10 weeks) was used to direct decisions about subsequent chemotherapy. For patients with response < 50% volume reduction, the protocol recommended switching to second-line chemotherapy regimens, including DOX (for patients in the SR group) or topotecan, carboplatin, CYC, and VP-16 for patients in the HR and VHR groups.

Local therapy consisted of primary or secondary resection and radiotherapy (RT). RT was recommended for all patients except for those with embryonal RMS (eRMS) and primary or secondary complete resection (IRSI, R0). A primary attempt at surgical resection was recommended only if it was considered possible without residuals (R0) and with avoidance of severe functional or cosmetic effects. Patients with primary or secondary R1 resection received adjuvant RT with 44.8 Gy. For patients in IRS group III, the decision on local therapy was made after three cycles of chemotherapy and assessment of response at weeks 7–10. For patients whose tumors were not amenable to an R0 resection, preoperative RT was recommended. RT dose was stratified according to histology and tumor response at week 7–10 and implemented at week 13–17. A total of 32 Gy were given to patients with eRMS and CR or GR; patients with eRMS and PR or aRMS and CR or GR received 44.8 Gy. Fractionation 2 × 1.6 Gy/day given at least 6 h apart (accelerated, hyperfractionated) was recommended for all RT fields except whole abdomen, neuroaxis, heart, liver, lung, and optic chiasma. The same principle was applied for decisions concerning the irradiation of the involved regional lymph nodes. For patients with OR and PD, the local therapy was modified and decided on an individual basis. DOX and ACTO-D were omitted during radiotherapy. The mandated cumulative DOX dose should, however, be completed after RT. Alternative techniques such as brachytherapy and proton therapy were permitted if clinically indicated. Supportive care recommendations were outlined in the protocol. Granulocyte colony-stimulating factor was not recommended prophylactically.

### Statistical Considerations

The primary outcomes were event-free survival (EFS) and overall survival (OS). EFS was defined as the time elapsed between the date of diagnosis and either the occurrence of an event or the date of the last patient contact. Event was defined as relapse of disease (local, metastatic, or combined) in patients who achieved complete remission, disease progression (defined as growth of tumor in patients who did not achieve complete remission), or death. Second malignancy was not defined as an event as described in the protocol. Overall survival (OS) was defined as the time from diagnosis to death or last follow-up for surviving patients.

Secondary objectives were to evaluate the modified risk stratification system; to assess the impact of reducing the cumulative dose of IFO in the subgroup A of the SR group (54 g/m^2^ to 24 g/m^2^); to assess the impact of an increase in DOX dose intensity, i.e., cumulative dose of 320 mg/m^2^ given in the first 16 weeks of chemotherapy; to examine the feasibility and impact on the prognosis of 25 weeks of maintenance therapy (MT) with CYC/VBL in patients in the HR and VHR groups who achieved CR (by any means) at the end of intensive systemic and local therapy.

The results obtained within the framework of this study were compared with the previous CWS studies and the EpSSG RMS 2005, conducted almost contemporarily since the risk stratification system was identical in both studies.

Data were prospectively collected and analyzed in the CWS international study center in Stuttgart. Data on adverse events were collected using the Word Health Organization coding system. EFS and OS were calculated using the Kaplan–Meier method. No formal interim monitoring of EFS and OS was planned, but the outcome was analyzed and presented at the annual study committee meetings. Univariate and multivariate hazard ratios and their 95% confidence intervals for the categories considered as potential prognostic were calculated with the corresponding Cox regression model, and significance testing was performed by the Wald-test. In addition, the 5 years EFS and OS were calculated for these categories extended by the individual primary sites and the fusion status. Differences between the corresponding EFS/OS-survival curves were evaluated by the log-rank test. The study protocol specified that factors used in the risk grouping would be assessed for their effect on EFS and OS. Statistical analyses were performed using SPSS statistics 22–25.0.0 (IBM Corporation, Armonk, NY, USA) and R 3.02 (Bell Laboratories, Murray Hill, NJ, USA) software packages.

## 3. Results

### 3.1. Patients Characteristics

Between 17.1.2003 and 31.12.2010, 498 patients were enrolled, 54 ineligible patients were excluded from this analysis (Figure 2), leaving 444 patients allocated to LR (23), SR (177), HR (219), or VHR (25) groups in the analytic cohort.

The patients’ characteristics are shown in Table 2. Twenty-eight patients with non-documented LN status in the institutional data sheets, who were allocated by their treating physician into SR and HR groups as N0, were regarded as N0 in the survival analysis. In 61 out of 81 aRMS, the fusion status was examined and found positive in 53 tumors (86% of the examined samples).

### 3.2. Outcome

An overview of the outcome is given in Table 3. A total of 94% of patients achieved complete remission, there were 120 treatment failures (27%), and the median follow-up was 9.6 years (IQR 7.6–10.9). The events were distributed as follows: isolated local failure 68 patients (15%), isolated distant failure 9 patients (2%), combined failure 8 patients (2%), primary progression (without achieving a CR) 31 patients (7%), unknown site of recurrence 4 patients (1%). The median time to relapse was 1.5 years (0.4–9). A total of 67% of relapses occurred within 2 years of diagnosis.

The tumor response, assessed after three cycles of chemotherapy in 212 cases with macroscopic tumors, was CR 6%, GR 58%, PR 22%, OR 12%, and PD 2%. Seven patients in the SR and 10 patients in the HR group with a tumor reduction < 50% switched to second-line treatment. The 5-year EFS and OS for the whole group were 73% (95% CI 69–77) and 80% (95% CI 76–84), respectively (Figure 3).

The 5-year EFS by risk group was 100% in the LR, 79% (95% CI 72–84) in the SR, 69% (95% CI 63–75) in the HR and 42% (95% CI 23–61) in the VHR (log-rank *p* = 0.000). The 5-year OS was 100%, 88% (95% CI 83–93), 76% (95% CI 70–82) and 42% (95% CI 23–61) respectively (log-rank *p* = 0.000), (Table 3, Figure 4). The 5-year EFS and OS in the SR-A vs. B subgroup was 81% (95% CI 72–90) vs. 77% (95% CI 69–85) (log-rank *p* = 0.34) and 96% (95% CI 92–100) vs. 83% (95% CI 76–90) (log-rank *p* = 0.04) respectively. In comparison, 10-year EFS and OS differ only minimally (Table 3) but single events occurred also after 5 years.

A total of 155 patients in the HR group, out of 204 who achieved CR, received MT at the end of intensive multimodal therapy, while 49 did not, based on the decision of the treating physician. The distribution of clinical variables in both groups did not differ, as shown (Appendix A). The 5-year EFS was 77% (95% CI 70–84%) for patients who received MT versus 63% (95% CI 50–76) for patients who did not (log-rank *p* = 0.015) and the 5-year OS was 84% (95% CI 78–90) versus 73% (95% CI 61–85) (log-rank *p* = 0.099) (Figure 5). In the VHR group, 16 patients out of 18 who achieved CR received MT (10 of whom were alive), and 2 did not (one was alive).

Radiotherapy was given to 82 (46%) patients in SR, 147 (67%) in HR, and 22 (88%) in the VHR group. A total of 90 patients died, 85 deaths were related to tumor relapse or progression.

There was one death from a toxic event (septic shock after surgery) during protocol therapy. Two patients died of non-protocol treatment-related toxicity, one from kidney failure after salvage chemotherapy and another as a result of liver failure after surgery of the relapse tumor. One patient succumbed to a glioblastoma occurring as a second tumor and one due to underlying genetic disorders. There were 22 (5%) second malignancies (SMN), 3 hematological and 19 solid tumors reported, in the SR group 13 (7.3%) and in the HR group 9 (4.1%). Details concerning SM are shown (Appendix A). The median time to second malignancy was 6 years (0.2–12) from the start of therapy. One patient developed a neuroblastoma almost simultaneously with the RMS. Fourteen patients out of 251 who were irradiated during the first treatment developed SM (5.5%), and 8/193 who were not (4.1%).

### 3.3. Toxicities

The most common grade 3–4 adverse events (using the World Health Organization coding system (WHO)), summarized across all courses of therapy were as follows in the four risk groups:

LR group leucopenia 4%, thrombocytopenia 4%;

SR group leucopenia 67%, thrombocytopenia 20%, anemia 3%, hepatopathy 4%, gastrointestinal toxicity 5%, clinically documented infection 10%, nephropathy 2%, peripheral neuropathy 1%;

HR group leucopenia 64%, thrombocytopenia 39%, anemia 7%, hepatopathy 1%, gastrointestinal toxicity 10%, clinically documented infection 16%, peripheral neuropathy 3%, central nervous toxicity 1%;

VHR group leucopenia 84%, thrombocytopenia 44%, anemia 4%, hepatopathy 4%, gastrointestinal toxicity 28%, clinically documented infection 12%, peripheral neuropathy 4%, central nervous toxicity 4%, skin toxicity 1%.

The organ dysfunction documented at the end of the multimodal therapy was: symptomatic cardiac dysfunction in the LR group 4%, SR group 2%, HR group 2%, and VHR group 4%. Renal dysfunction in the LR group 4%, SR group 16%, HR group 14%, and VHR group 8%. Endocrine disorders in the SR group 2%, HR group 3%, and VHR group 4%. Growth disorders in the SR group 3% and HR group 5%. Neuropathy in the SR group 6%, HR group 4%, and VHR group 8% (all of these were peripheral with the exception of 1 patient in the SR group who developed a central neuropathy).

### 3.4. Univariate and Multivariate Analysis

The following clinical variables used in the risk stratification were considered potential prognostic factors and included in the univariate and multivariate Cox regression analyses: age (≤10 vs. >10 years), size (≤5 vs. >5 cm), IRS stage, histology (unfavorable: aRMS vs. favorable: eRMS), primary site (favorable: GU-BP, HN-nPM, ORB vs. unfavorable: EXT, OTHER, HN-PM, GU-nBP), T-Status (T1 vs. T2) and N-Status (N0 vs. N1). Univariate hazard ratios for EFS and OS were significantly influenced by all variables except age (Appendix A). In the multivariate analysis significant negative prognostic factors for EFS were; IRS III vs. IRS I (HR 2.81, *p* = 0.02), T-Status T2 vs. T1 (HR 1.55 *p* = 0.02) and N-Status N1 vs. N0 (HR 1.57 *p* = 0.04) and for OS unfavorable histology (HR 1.79 *p* = 0.002), T-Status (HR 1.81 *p* = 0.02) and N-Status (HR 1.82 *p* = 0.02) (Appendix A). In addition, a univariate analysis was performed, which included not only primary sites by groups (favorable and unfavorable) but the seven single primary sites categories and fusion status in patients with aRMS (Appendix A). No clear grouping in the favorable and unfavorable groups has emerged, but the 5-year EFS rates of single primary sites decrease gradually from 84% for UG-nBP, 83% for ORB, 77% for UG-BP, 72% for HN-nPM, 70% EXT, 65% for HN-PM to 62% for OTH. aRMS histology, with or without fusion status, was highly associated with prognosis.

## 4. Discussion

One of the most pertinent aims in the evolution of therapy concepts for children with malignancies is the precise risk adaptation of the known effective therapy modalities with the goal to achieve the best possible outcome with as little therapy as possible. Treating young patients with cancer aims not only at achieving cure but at avoiding late effects such as permanent disabilities, second malignancies, organ dysfunction, and infertility. The objectives of the CWS-2002P study pursued this strategy. The risk stratification systems of the CWS-2002P and RMS 2005 studies, which included tumor size and patient`s age additionally to the risk factors included in previous European studies, were identical since they were based on common analyses and agreements between the CWS, AIEOP-STSC, and MMT groups. By including tumor size and patient age into risk stratification, patients from the previous HR group (CWS-96) who were <10 years with eRMS in the IRS group II and III were classified in the SR group despite unfavorable tumor site. The risk-adapted chemotherapy in the CWS-2002P without DOX in the LR and SR and without IFO in the LR and dose-reduced in the SR subgroup A (from 54 to 24 g/m^2^, CYC equivalent dose approximately 13.5 g/m^2^ to 6 g/m^2^) did not impair EFS and OS as compared to previous CWS and other published results [1,7,8,10,11]. Some patients treated in our SR group were treated on the COG protocol ARST0331 in the LR group with a very low cumulative dose of CYC (4,8 g/m^2^) without a compromised outcome in subset 1 but with inferior EFS in subset 2, indicating an important role of CYC dose for prognosis [16,17]. Reduction in the CYC dose in the COG ARST0531 trial on the intermediate-risk group (to 8.4–16.8 g/m^2^ from 25.1 to 30.8 g/m^2^), which is partially comparable to the HR and VHR groups in this trial, was discussed as one of the reasons for the poorer outcome compared with the previous trial D9803 [18]. The cumulative dose of IFO in the HR and VHR group in the CWS-2002P was 54 g/m^2^ (CYC equivalent dose approximately 13.5 g/m^2^), which is similar to the CYC dose in the ARST0531 trial, without increased therapy failure rates. It is, therefore, still not clear what is the optimal dose of alkylating agent and whether the dose per cycle, dose intensity, or cumulative dose are crucial. The outcome analysis by single primary tumor sites demonstrated that the grouping into favorable and unfavorable sites, which was based on the common agreement of the CWS, AIEOP-STSC, and MMT study groups and used in the CWS-2002P study, the EpSSG RMS 2005 and the CWS-Guidance have to be reevaluated. In the multivariate analysis, they had no influence on EFS or OS. The GU-BP, which is regarded as an unfavorable site, had a very good outcome (EFS 78% (95% CI 67.3–90.4)) comparable to other favorable sites ORB, HN-nPM, and GU-nPB [19]. Similar to the RMS 2005 analysis of risk stratification, the GU-BP site had an EFS of 75.8% (65.6–83.4) and is regarded as a favorable site in the ongoing FaR-RMS EpSSG trial [11]. The distribution of histology in alveolar (*n* = 74) vs. non-alveolar (*n* = 355) compared to “fusion status positive (*n* = 50) or negative” did not yield a major difference in terms of prognostic significance since only six tumors with alveolar histology were “fusion negative”. Other features such as histological anaplasia (often associated with TP53 mutations) or spindle cell morphology (associated with MYOD1 mutation or VGLL2 fusions) have been shown to be relevant for prognosis but were not systematically recorded in the CWS-2002P database, so we could not analyze their impact on the outcome.

A further aim was to clarify the possible significance of DOX dose intensity. The role of DOX in the treatment of patients with RMS has been controversial for over 40 years [1,20] despite many trials both in North America and Europe investigating this question. The comparisons of results between groups were complicated due to differences in risk assignment. Two European studies investigated the role of therapy intensification in a randomized way, comparing three-drugs regimen IVA (MMT95, 1995–2003) and four-drugs regimen IVA plus DOX cumulative dose 240 mg/m^2^ (CWS/ICG-96, 1996–2003) with a six-drugs regimen CEVAIE (epirubicine instead of DOX, cumulative dose 450 mg/m^2^, plus etoposide and carboplatin) in the identical defined HR group showing no difference in outcome between therapy arms group [8,9]. The EpSSG RMS 2005 study, conducted between 2005 and 2013, randomized the IVA regimen vs. IVA plus DOX (given simultaneously with ACTO-D in the first four cycles, (11 weeks, cumulative dose 240 mg/m^2^)) and again showed no improvement in the outcome of patients with high-risk RMS^10^. The CWS-2002P study conducted almost contemporarily (2003–2010) pursued a similar question of the DOX dose intensity for the HR group but with an increased cumulative dose of 320 mg/m^2^ given in the first 16 weeks and achieved identical EFS as RMS 2005. Similarly, no improvement of prognosis for the HR group was achieved when compared to the CWS-96 study (standard arm: DOX cumulative dose of 240 mg/m^2^ or experimental arm: epirubicine 450 mg/m^2^ applied during the first 22 weeks) [8]. The CWS Group, therefore, recommended IVA/VA as standard therapy for patients with SR and HR localized RMS in the CWS-Guidance, which outlines therapy recommendations for localized RMS since 2010. CWS studies have previously evaluated the role of low-dose chemotherapy given at the end of the standard therapy as an MT and showed promising effects in patients with metastatic RMS [21]. The CWS-2002P study investigated in a non-randomized way the feasibility, tolerability, and impact on the prognosis of 6-months low-dose cyclophosphamide and vinblastine in patients in HR and VHR groups and showed that this therapy was feasible, well tolerated, and improved EFS. RMS 2005 investigated the role of a similar MT (vinorelbine instead of vinblastine) by randomizing HR patients who have received 25 weeks of IVA+/− doxorubicin to receive or not receive an additional 24 weeks MT. The MT seemed to improve survival but not disease-free survival [22]. One has to keep in mind that one of the main reasons to use IFO in the standard RMS treatment in Europe instead of CYC as used in the USA, was the assumed lower gonadal toxicity. Cumulative doses of CYC in the MT (CWS-2002P 7.35 and RMS 2005 4.2 g/m^2^) are relevant for gonadal toxicity, which has to be considered in assessing the late effects of the MT. The role of MT continues to be evaluated. The CWS group is now conducting the CWS-2007HR study (EudraCT number: 2007-001478-10) in which patients with RMS and RMS-like tumors in HR and VHR are randomly assigned to receive another MT with etoposide, idarubicin, and trofosfamide following standard IVA or VAIA therapy. The new EpSSG FaR-RMS trial (EudraCT number: 2018-000515-24) investigates the role of a longer duration of MT with cyclophosphamide and vinorelbine (randomization 6 vs. 12 months).

## 5. Conclusions

The study shows that a reduction in the IFO dose did not deteriorate the outcome, and an increase in the dose intensity of doxorubicin did not improve prognosis in comparison to the CWS-91 and -96 and EpSSG RMS 2005 study. This is an important message in relation to the acute and long-term effects of therapy. The maintenance therapy with cyclophosphamide and vinblastine applied at the end of multimodal therapy in the high- and very high-risk groups improved event-free survival and has to be further evaluated, both regarding the outcome and long-term sequelae. This provides additional information to the results of the RMS 2005 study that investigated vinorelbine in MT. Vinblastine is more easily accessible in many countries and could be used as an alternative to vinorelbine. Our analysis also shows that the established assignment of primary sites into favorable or unfavorable, based on the consensus of the European and North American cooperative soft tissue sarcoma groups, needs to be revised. In our opinion, the current risk stratification system in general, which was historically developed through joint analyzes of potential risk factors within the framework of international cooperation, should be reassessed. An international effort is needed before relevant biologic subtypes may eventually be incorporated into standard practice [23]. The first step to harmonize approaches to optimize RMS risk stratification a collaboration of the three cooperative groups: Children’s Oncology Group (COG), Cooperative Weichteilsarkom Studiengruppe (CWS), and European pediatric Soft tissue sarcoma Study Group (EpSSG) was initiated, resulting in the INternational Soft Tissue SaRcoma ConsorTium (INSTRuCT) (https://commons.cri.uchicago.edu/instruct/ accessed on 8 February 2022).

## Figures and Tables

**Figure 1 cancers-14-00899-f001:**
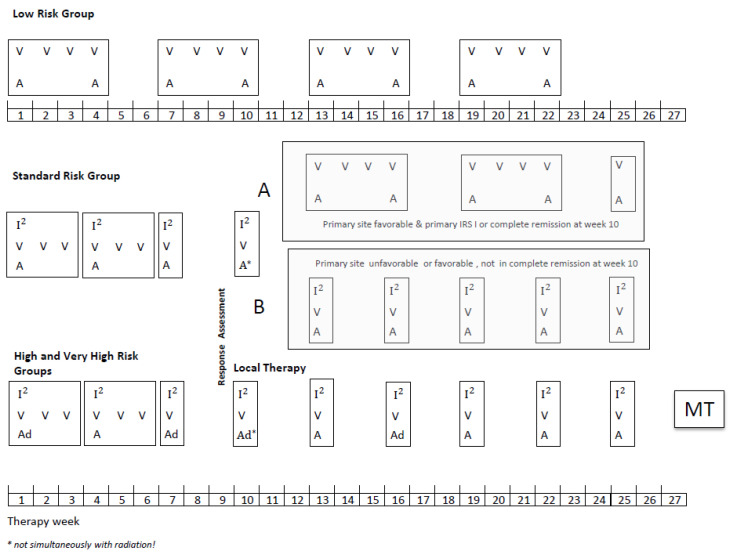
CWS-2002P treatment plan.

**Figure 2 cancers-14-00899-f002:**
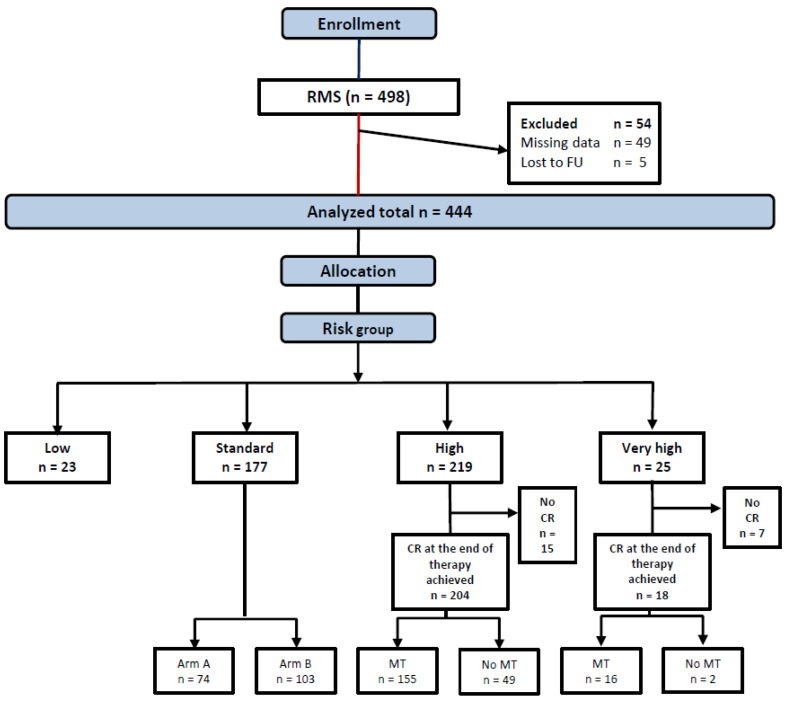
Consort diagram.

**Figure 3 cancers-14-00899-f003:**
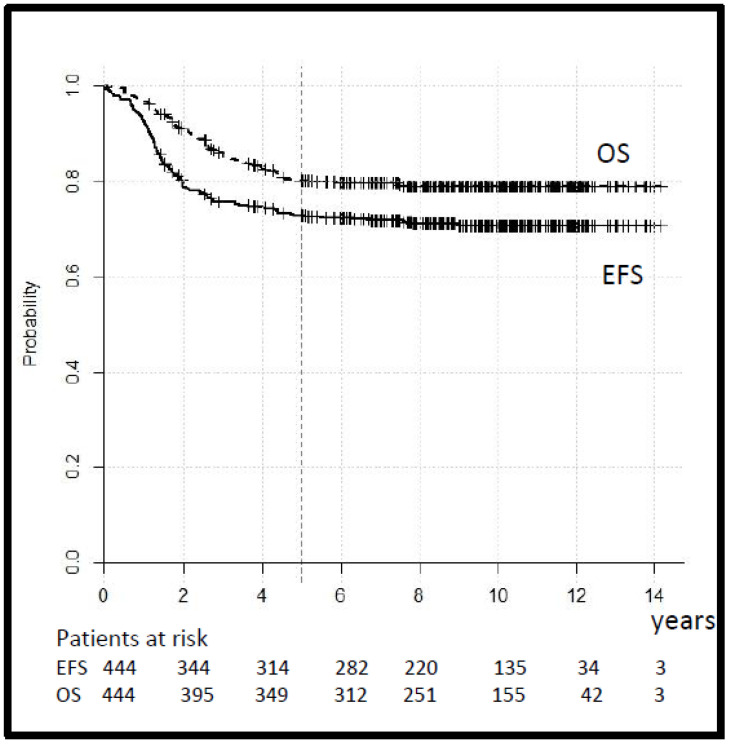
Legend: event-free and overall survival (EFS and OS) for all patients.

**Figure 4 cancers-14-00899-f004:**
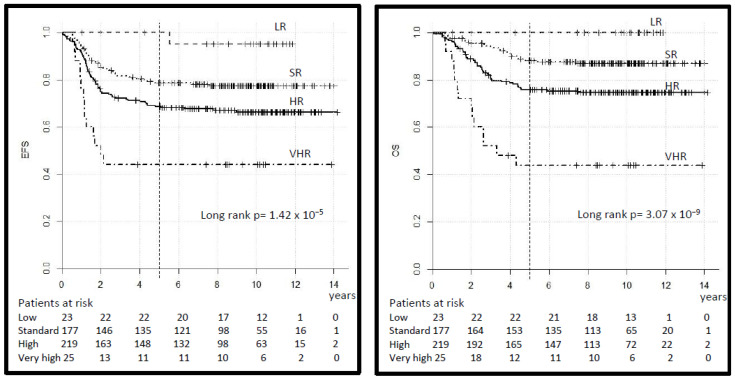
Legend: event-free and overall survival (EFS and OS) by risk group.

**Figure 5 cancers-14-00899-f005:**
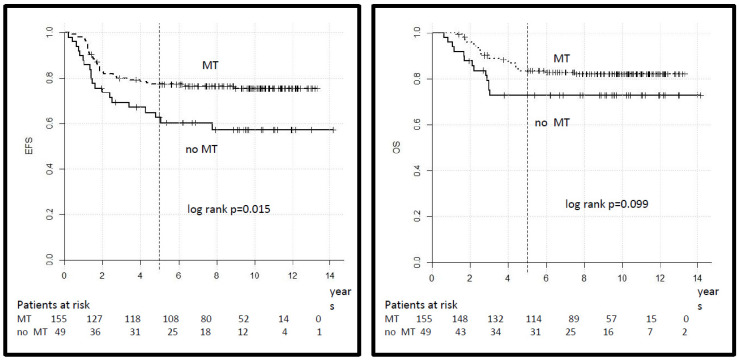
Legend: event-free and overall survival for patients in the HR group according to treatment with maintenance therapy (MT).

**Table 1 cancers-14-00899-t001:** Risk stratification for patients with localized rhabdomyosarcoma.

Risk Group	Pathology	IRS Group	Site	LN Stage	Size and Age
Low (LR)	eRMS	I	Any	N0	Favorable
Standard (SR)	eRMS	I	Any	N0	Unfavorable
	eRMS	II, III	Favorable	N0	Any
	eRMS	II, III	Unfavorable	N0	Favorable
High (HR)	eRMS	II, III	Unfavorable	N0	Unfavorable
	eRMS	II, III	Any	N1	Any
	aRMS	I, II, III	Any	N0	Any
Very High (VHR)	aRMS	I, II, III	Any	N1	Any

Abbreviations and/or definitions: IRS group—postsurgical stage; Pathology: eRMS—all embryonal RMS spindle cell and botryoid, aRMS–all alveolar RMS (including solid-alveolar variant); Site—primary tumor site Favorable: orbit (ORB), genitourinary non-bladder and prostate GU-NBP (i.e., paratesticular, vagina/uterus), head and neck non-paramenigeal (HN-NPM), Site—primary tumor site Unfavorable: head and neck paramenigeal (HN-PM), genitourinary bladder and prostate (GU-BP), extremities (EXT), “other site” (OTH); LN Stage—regional lymph node status; Size and Age Favorable: ≤5 cm and ≤10 years, Size and Age Unfavorable: >10 years and/or >5 cm.

**Table 2 cancers-14-00899-t002:** Patient characteristics.

			Risk Group							
	Total		Low		Standard		High		Very High	
	No.	%	No.	%	No.	%	No.	%	No.	%
	444	100	23	5	117	40	219	49	25	6
**Variable**										
Sex										
Male	260	58	21	91	103	58	123	56	13	54
Female	184	42	2	9	74	42	96	44	12	46
**Age**										
≤10 years	331	74	23	100	135	76	162	74	11	46
>10 years	113	25	-	-	42	24	57	26	14	54
**Histology**										
aRMS fusion positive	53	12	-	-	-	-	34	16	19	76
aRMS fusion negative	7	2	-	-	-	-	7	4	-	-
aRMS fusion unknown	21	4	-		-	-	14	7	6	31
Non-aRMS	363	82	23	100	177	100	163	74	-	-
**Tumor site**										
EXT	38	9	-	-	4	2	30	14	4	15
HN-nPM	45	10	2	9	22	12	19	9	2	8
HN-PM	114	26	-	-	26	15	76	-	12	46
ORBITA	44	10	-	-	41	23	3	1	-	-
GU-BP	50	11	-	-	21	12	29	13	-	4
GU-nBP	80	18	21	91	53	30	6	3	-	-
OTH	72	16	-	-	10	6	55	-	7	27
Not specified	1		-	-	-	-	1	-	-	-
**Tumor size**										
≤5 cm	223	50	23	100	132	74	62	28	6	23
>5 cm	206	47	-	-	42	24	145	66	19	77
Not specified	15	3	-	-	3	2	12	6	-	-
**IRSG**										
I	55	12	23	100	27	15	5	2	-	-
II	67	15	-	-	43	24	23	11	1	4
III	322	73	-	-	107	61	191	87	24	96
**Tumor invasiveness**										
T1	210	47	19	83	115	65	70	32	6	27
T2	204	46	3	13	54	31	129	59	18	69
TX	30	7	1	4	8	4	20	9	1	4
**Regional lymph nodes**										
N0	350	79	23	100	169	96	158	73	-	-
N1	66	15	-	-	-	-	41	18	25	100
NX	28	6	-	-	8	4	20	9	-	-

**Table 3 cancers-14-00899-t003:** Failures and survival parameters by risk groups.

	Risk Group				
	Low *n* = 23	Standard *n* = 177	High *n* = 219	Very High *n* = 25	Total *n* = 444
	*n* (%)	*n* (%)	*n* (%)	*n* (%)	*n* (%)
**1st Complete remission (CR) achieved**	23 (100)	172 (97)	204 (93)	18 (73)	417 (94)
**Failures**					
Local	1 (4)	26 (15)	37 (17)	4 (16)	68 (15)
Metastatic	-	-	6 (3)	3 (12)	9 (2)
Combined	-	3 (2)	5 (2)	-	8 (2)
Progression	-	6 (3)	18 (8)	7 (28)	31 (7)
Not specified	-	1 ()	3 (1)	-	4 (1)
**Total failures**	1 (4)	36 (20)	69 (32)	14 (56)	120 (27)
**Sec malignancy**	0	13	9	0	22
Alive	23 (100)	155 (88)	165 (75)	11 (44)	354 (80)
Dead	0	22 (12)	54 (25)	14 (56)	90 (20)
DOD	0	20	51	14	85
DOT	0	0	1	0	1
DoOT	0	0	2	0	2
DOC	0	2	0	0	2
**Median follow-up** years (IQR)	9.8 (7.8–10.6)	9.9 (7.7–10.6)	9.7 (7.4–11.3)	9.3 (8.4–10.2)	9.6 (7.6–10.9)
**EFS** 5 years rate% (95% CI)	100	79 (72–84)	69 (63–75)	42 (23–61)	73 (69–77)
**EFS** 10 years rate% (95% CI)	95 (72–84)	77 (70–84)	67 (71–83)	42 (23–61)	71 (67–75)
**OS** 5 years rate% (95% CI)	100	88 (83–93)	76 (70–82)	42 (23–61)	80 (76–84)
**OS** 10 years rate% (95% CI)	100	87 (84–90)	75 (69–81)	42 (23–61)	79 (75–83)

Legend: CR = complete remission, DOD = dead of disease, DOT = dead of therapy, DOC = dead of other causes, DOoT = dead of other therapy (non-CWS) EFS = event-free survival. FU = median follow-up (in years) (min-max) for patients alive, LN = lymph node, OS= overall survival, RT = radiotherapy, ukn = unknown.

## Data Availability

Individual participant data are not publicly available since this requirement was not anticipated in the study protocol.

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
