# Peer review of "Long-Term Clinical Outcome and Prognostic Factors of Children and Adolescents with Localized Rhabdomyosarcoma Treated on the CWS-2002P Protocol"

_cancers, 2022, doi:10.3390/cancers14040899_

Round 1

Reviewer 1 Report

This manuscript is a well written description of the results of the CWS-2002P trial of localized rhabdomyosarcoma. Although it mainly replicates known features of risk in these patients, the study does provide some additional data on the cause of failure in these patients as well as the role of maintenance chemotherapy. The conclusion to re-evaluate risk stratification using molecular features is appropriate.

Some minor edits may improve the manuscript.

  1. I wonder if additional pathology comments might be added to a supplemental table. Including comments on anaplasia and spindle morphology from the central path review would add value given recent findings that these features may mark biologically more aggressive tumors.
  2. There is little mention of the potential confounding variable of radiation dose 32 vs 44 Gy. This should be included in the multivariate analysis.
  3. Language about maintenance chemotherapy should be tempered in the abstract and clarify that these findings should be studied in a well controlled prospective trial of this patient population.

Author Response

Reviewer: 1

This manuscript is a well written description of the results of the CWS-2002P trial of localized rhabdomyosarcoma. Although it mainly replicates known features of risk in these patients, the study does provide some additional data on the cause of failure in these patients as well as the role of maintenance chemotherapy. The conclusion to re-evaluate risk stratification using molecular features is appropriate.

Some minor edits may improve the manuscript.

  1. I wonder if additional pathology comments might be added to a supplemental table. Including comments on anaplasia and spindle morphology from the central path review would add value given recent findings that these features may mark biologically more aggressive tumors.

Response: The reviewer's comment addresses an important aspect of the biological assessment of aggressive tumors. In the Risk Grouping system used in the CWS-2002P  (identical with the EpSSG RMS 2005) only two histological  subgroups were used: alveolar – unfavorable and embryonal  (non-alveolar) – favorable. RMS with diffuse anaplasia and spindle cell RMS (later identified as having MyOD1 mutation in older patients) were recognized (WHO 2002) as possible unfavorable factors but were not used in our stratification system and therefore not documented in the data base. We are unable therefore to comment on these subtypes based on our analysis

Additional aspect is that each cooperative group (COG, EpSSG and CWS) currently assesses anaplasia differently, making comparisons of prognostic relevance difficult. It is not clear how to incorporate this into routine prognostic testing, and more work is needed to determine how to best identify this group as well as the relationship of TP53 germline mutations and histologic anaplasia.

The current EpSSG FaRMS study replaced histology by fusion status (negative and positive) and has not included any other pathologic or molecular subgroups in the risk grouping.  In contrast the COG group based on the publication of J. Shern (JCO 2021) introduced MyOD1 mutation as a negative prognostic factor. Still an international effort is needed before relevant biologic subtypes  may eventually be incorportated into standard practice. (Rudzinski et al.  Pediatr Blood Cancer. 2020;e28798., https://doi.org/10.1002/pbc.28798). The INternational Soft Tissue SaRcoma ConsorTium (INSTRuCT) Tissue SaRcoma ConsorTium (INSTRuCT) (https://commons.cri.uchicago.edu/instruct/) composed of the three cooperative clinical trials organizations: Children's Oncology Group (COG), Cooperative Weichteilsarkom Studiengruppe (CWS), and European paediatric Soft tissue sarcoma Study Group (EpSSG) will maximize potential to optimize risk stratification. (Hawkins et al. Pediatr Blood Cancer. 2020;e28701., https://doi.org/10.1002/pbc.28701)

We added in the discussion a sentence addressing this issue as suggested by the reviewer::

“Other features such as histological anaplasia (often associated with TP53 mutations) or spindle cell morphology (associated with MYOD1 mutation or VGLL2 fusions) have been shown to be relevant for prognosis but were not systematically recorded in the CWS 2002P database, so we could not analyze their impact on outcome.”

  1. There is little mention of the potential confounding variable of radiation dose 32 vs 44 Gy. This should be included in the multivariate analysis.

Response: The 32-Gy and 44.5-Gy doses were stratified according to various criteria such as response to preoperative treatment, chemotherapy, and histological subtype. We included only patients and tumor-related factors in the multivariate analysis to reassess the risk stratification group, but not the therapy variables. Local therapy was not the main objective of this analysis and will be analyzed separately. Therefore, we believe that including radiotherapy doses in the multivariate analysis, along with other non-therapy-related factors, could lead to results that cannot be properly interpreted (comparisons of 32Gy vs. 44.5 Gy) since only patients with very good prognostic factors were irradiated with 32 Gy. (Koscielniak et al. JCO 2014;32 (Issue 15S):645s)

  1. Language about maintenance chemotherapy should be tempered in the abstract and clarify that these findings should be studied in a well controlled prospective trial of this patient population.

Response: We have taken into account the reviewer's important note on the interpretation of maintenance therapy in the abstract. We thought that the formulation  "seems to improve prognosis "  was sufficiently careful. Following the reviewer`s suggestion however, we have changed it into:

 “MT was feasible, seemed to have impact on prognosis and should be studied in a well controlled prospective trial in this patient population.”

Reviewer 2 Report

This well-written and important paper summarizes the findings of iteratively-developed German study CWS-2002P, focusing on pediatric localized rhabdomyosarcoma patients on study from 2003 to 2010. The study was well-done with much valuable data generated for future study design and to establish the current European standard of care. The authors are to be commended for their efforts.

Should there be any question about the preponderance of localized PM disease in the H&N patients, the authors can refer readers to PMID: 10491544, which described older studies IRS-II and IRS-III, where 68% were cranial PM.

Minor questions arose while reviewing the paper:

  1. It seems that if confidence intervals for survival (EFS) with or without maintenance therapy (MT) are overlapping, it would be difficult if not impossible to achieve statistical significance. Could the authors clarify on this?
  2. Given significance of T-status in this study, will this be re-incorporated into future study risk stratification (Table S1)? Could the authors please discuss why T-status was removed from stratification?
  3. Given the rate of secondary malignancies is identical to other series where G-CSF is used, the protocol mandate against growth factor should be discussed. Could the authors please address this? Were there any issues with treatment delay given the high rates of bone marrow toxicity documented in SR, HR, and VHR groups? Could this have affected outcome?
  4. Lines 353-354 discuss aRMS with or without fusion status - please clarify this statement with the number (N) in each category. The low N makes this conclusion, which is contrary to established data, difficult to state conclusively and this should be made clear. The concluding line on line 453 also appears to make a conclusion out of this limited data, which should be adjusted or eliminated. It is unclear what "molecular-based risk parameters" the authors are suggesting would be useful and clarification is necessary. 
  5. The commentary on primary site grouping suggests making GU-BP favorable prognosis with discussion of available data to support this but appears to neglect discussing making HN-nPM unfavorable prognosis (Table S7). Could this latter point please be discussed further and if the literature supports this categorization. Furthermore, smaller p-values beyond the threshold of significance do not imply "more" significance and caution should be applied to interpreting this post-hoc evaluation of groupings in which multiple proposed groupings, including classical, are all significant.

Author Response

Reviewer 2

This well-written and important paper summarizes the findings of iteratively-developed German study CWS-2002P, focusing on pediatric localized rhabdomyosarcoma patients on study from 2003 to 2010. The study was well-done with much valuable data generated for future study design and to establish the current European standard of care. The authors are to be commended for their efforts.

Should there be any question about the preponderance of localized PM disease in the H&N patients, the authors can refer readers to PMID: 10491544, which described older studies IRS-II and IRS-III, where 68% were cranial PM.

Minor questions arose while reviewing the paper:

  1. It seems that if confidence intervals for survival (EFS) with or without maintenance therapy (MT) are overlapping, it would be difficult if not impossible to achieve statistical significance. Could the authors clarify on this?

Response: If the confidence intervals partially overlap, it cannot be concluded that the parameters do not differ significantly. If the overlap is not too large (6%) and the number of patients not too small, the difference may still be significant. In our case resulting in a significant p of 0.015 (Source : Book Applied statistics L.Sachs u. J. Hedderich)

2. Given significance of T-status in this study, will this be re-incorporated into future study risk stratification (Table S1)? Could the authors please discuss why T-status was removed from stratification?

Response: Facing this interesting point of view it has to be noticed that  T was a risk factor in the risk grouping of the CWS-96 study based on a European consensus between the study groups CWS, the Italian ICG STS Group and the MMT SIOP (and also in the ICG RMS-96 study and the MMT95 study has been used). A follow-up analysis performed on the CWS/RMS-96 data and validated on data from studies with longer follow-up: the SIOP-MMT-84 and -89 studies, the German CWS-81 - and -91 studies and the Italian RMS-79 and RMS-88 studies identified as significant: histology, IRS group, tumor location and size, N involvement, and age > 10 years, but not T status. The combination of these factors determined the risk grouping used in the CWS-2002P and EpSSG RMS2005. This grouping system is also used in the CWS guideline. The new EpSSG FaRMS trial (the CWS group will participate in the study) continues to use a similar risk grouping system without T status. In addition, the assignment of T1 and T2 by the local investigators was often incorrect (after a central assessment). It is therefore difficult to use it in a large international study. Nevertheless, we documented and centrally validated the T status in the CWS 2002P, and were able to incorporate it into the risk factor analysis. In summary, the decision to delete T status was a consensus decision, facilitating comparison between studies and reducing potential errors. A new, international risk grouping  for RMS system is  one of the goals of the INternational Soft TissueSaRcoma ConsorTium (INSTRUCT) composed of three large cooperative clinical trials organizations: Tissue

SaRcoma ConsorTium (INSTRuCT) (https://commons.cri.uchicago.edu/instruct/) Children's Oncology Group (COG), Cooperative Weichteilsarkom Studiengruppe (CWS), and European paediatric Soft tissue sarcoma Study Group (EpSSG) to replace competing systems used in Europe and North America.( Hawkins, Bisogno, Koscielniak. Pediatr Blood Cancer. 2020;e28701.https://doi.org/10.1002/pbc.28701)

3. Given the rate of secondary malignancies is identical to other series where G-CSF is used, the protocol mandate against growth factor should be discussed. Could the authors please address this? Were there any issues with treatment delay given the high rates of bone marrow toxicity documented in SR, HR, and VHR groups? Could this have affected outcome?

Response: SMN in patients with RMS and other STS can be associated with genetic predisposition, cumulative alkylators and anthracyclines doses, irradiation and also the use of G-CSF. (Caruso et al. PBC 2019) In our series, only 2 hematologic malignancies occurred (0.5%) that are thought to be associated with the prophylactic use of G-CSF (Lyman et al. Ann Oncol 2018) ). Bisogno et al. (Lancet Oncology 2018)  reported 2% ( IVA arm) and 4% ( IVA plus Doxo arm) SMN in the RMS2005 study but there is no information concerning the type of the SMN. The main reason for not using the G-CSF prophylactically was the lack of clinical need, based on our experience in the previous studies. A leucopenia Grade III-IV(1000-1900/µl) is not always life-threatening and is not always associated with severe neutropenia (<500/µl).  The rate of clinically documented infection was much lower (10-16%).  The use of G-CSF was recommended in clinically documented infection with neutropenia. We have not seen major treatment delays due to leucopenia that occurred mainly in the treatment interval. Since the topic concerning the indications for the prophylactic use of G-CSF, its possible impact on the outcome and the rate of second malignancies is very complex, we think that it would be would go beyond the scope of this analysis to discuss  all these aspects in this article.

4. Lines 353-354 discuss aRMS with or without fusion status - please clarify this statement with the number (N) in each category. The low N makes this conclusion, which is contrary to established data, difficult to state conclusively and this should be made clear. The concluding line on line 453 also appears to make a conclusion out of this limited data, which should be adjusted or eliminated. It is unclear what "molecular-based risk parameters" the authors are suggesting would be useful and clarification is necessary. 

Response: In line with the reviewer's advice, we have clarified the information on aRMS with or without fusion status. As recommended, we have added the numbers of patients in each category: (probably lines 374-376 were meant).

“The distribution of histology in alveolar (n=74) vs. non-alveolar (n=355) compared to "fusion status positive (n=50) or negative (n=6) " did not yield a major difference in terms of prognostic significance since only six tumors with alveolar histology were "fusion negative".

We do not think that our data contradicts other published data. We only found that fusion status vs. pathology in the univariate analysis did not cause any fundamental change in EFS and OS (histology unfavorable n=74, 5-year EFS 59%, favorable n=355 5-year EFS 75%, Fusion positive n=50 5-year EFS 54% , Fusion negative (aRMS) n=6 5-year EFS 100%, Fusion unknown EFS 5-year 61% ) and the  fusion status was not a stronger predictor of poor outcome. The difference of our analysis to the published data like Hibbitts et al. Cancer Medicine 2019, is that by “fusion negative” we mean only the fusion negative alveolar RMS. Of course, the embryonal RMS are also probably fusion negative (although not all tested), but we wanted to examine the group of “fusion negative” that were diagnosed by our reference pathologists as clearly alveolar separately and not mix these 6 patients with the 355 embryonal RMS.

The diagnostic criteria for alveolar RMS used by our reference pathologists were different in comparisons to the MMT SIOP, EpSSG and COG groups and the rate of fusion negative aRMS was very low in our series. Since the pathology further plays an important diagnostic role, we think that it should be still taken into account.

Since only the most frequent fusion like Foxo1/Pax3/7 are routinely tested, it is still possible that other rare fusions like NCOA1,2/Pax3/7, PAX3-FOXO4,  FOXO1-FGFR1 might be present. Genomic analyses of RMS tumors show that other features, specifically TP53 m or MYOD1 mutations, may influence outcome. (Rudzinski et al. PBC 2020,Pediatr Blood Cancer) 2020;e28798. https://doi.org/10.1002/pbc.28798.   TP53 and MYOD1 Mutations present mostly in “fusion-negative” RMS but associated with very poor prognosis have been already included in the risk grouping by the COG. We therefore think that the reducing the RMS classification into "Foxo Fusion-positive vs. negative" has to be probably extended with other prognostic relevant markers, taking into account the deep heterogeneity of embryonal RMS. Missiaglia et al (JCO 2012) reported a very good prognosis of patients with Pax7/Foxo positive tumors, identical with fusion negative aRMS. There is therefore still an international effort needed to define, validate and introduce molecular markers into the classification of RMS. However, we agree with the reviewer that we cannot draw such a conclusion based on our present analysis. Since reviewer 1 recommended a comment concerning the anaplasia and spindle cell tumors, we have already added a statement on this.

We have adjusted the conclusion – as suggested by the reviewer.

“ In our opinion, the current risk stratification system in gemneral, which was historically developed through joint analyzes of potential risk factors within the framework of international cooperation should be reassessed. An international effort is needed before relevant biologic subtypes  may eventually be incorportated into standard practice. The first step to harmonize approaches to optimize RMS risk stratification is a collaboration of the three cooperative groups: Children's Oncology Group (COG), Cooperative Weichteilsarkom Studiengruppe (CWS), and European paediatric Soft tissue sarcoma Study Group (EpSSG) was initiated resulting in the INternational Soft Tissue SaRcoma ConsorTium (INSTRuCT) (https://commons.cri.uchicago.edu/instruct/)”

5. The commentary on primary site grouping suggests making GU-BP favorable prognosis with discussion of available data to support this but appears to neglect discussing making HN-nPM unfavorable prognosis (Table S7). Could this latter point please be discussed further and if the literature supports this categorization. Furthermore, smaller p-values beyond the threshold of significance do not imply "more" significance and caution should be applied to interpreting this post-hoc evaluation of groupings in which multiple proposed groupings, including classical, are all significant.

Response: Our idea was to analyze the different grouping of primary sites, since the old grouping was based on the consensus reached many decades ago. However, we agree with the reviewer that p-values above the significance threshold do not imply "more significance" and we deleted Table S7. The new risk stratification must be based on more complex mathematical modeling of various parameters

Reviewer 2

This well-written and important paper summarizes the findings of iteratively-developed German study CWS-2002P, focusing on pediatric localized rhabdomyosarcoma patients on study from 2003 to 2010. The study was well-done with much valuable data generated for future study design and to establish the current European standard of care. The authors are to be commended for their efforts.

Should there be any question about the preponderance of localized PM disease in the H&N patients, the authors can refer readers to PMID: 10491544, which described older studies IRS-II and IRS-III, where 68% were cranial PM.

Minor questions arose while reviewing the paper:

  1. It seems that if confidence intervals for survival (EFS) with or without maintenance therapy (MT) are overlapping, it would be difficult if not impossible to achieve statistical significance. Could the authors clarify on this?

Response: If the confidence intervals partially overlap, it cannot be concluded that the parameters do not differ significantly. If the overlap is not too large (6%) and the number of patients not too small, the difference may still be significant. In our case resulting in a significant p of 0.015 (Source : Book Applied statistics L.Sachs u. J. Hedderich)

  1. Given significance of T-status in this study, will this be re-incorporated into future study risk stratification (Table S1)? Could the authors please discuss why T-status was removed from stratification?

Response: Facing this interesting point of view it has to be noticed that  T was a risk factor in the risk grouping of the CWS-96 study based on a European consensus between the study groups CWS, the Italian ICG STS Group and the MMT SIOP (and also in the ICG RMS-96 study and the MMT95 study has been used). A follow-up analysis performed on the CWS/RMS-96 data and validated on data from studies with longer follow-up: the SIOP-MMT-84 and -89 studies, the German CWS-81 - and -91 studies and the Italian RMS-79 and RMS-88 studies identified as significant: histology, IRS group, tumor location and size, N involvement, and age > 10 years, but not T status. The combination of these factors determined the risk grouping used in the CWS-2002P and EpSSG RMS2005. This grouping system is also used in the CWS guideline. The new EpSSG FaRMS trial (the CWS group will participate in the study) continues to use a similar risk grouping system without T status. In addition, the assignment of T1 and T2 by the local investigators was often incorrect (after a central assessment). It is therefore difficult to use it in a large international study. Nevertheless, we documented and centrally validated the T status in the CWS 2002P, and were able to incorporate it into the risk factor analysis. In summary, the decision to delete T status was a consensus decision, facilitating comparison between studies and reducing potential errors. A new, international risk grouping  for RMS system is  one of the goals of the INternational Soft TissueSaRcoma ConsorTium (INSTRUCT) composed of three large cooperative clinical trials organizations: Tissue

SaRcoma ConsorTium (INSTRuCT) (https://commons.cri.uchicago.edu/instruct/) Children's Oncology Group (COG), Cooperative Weichteilsarkom Studiengruppe (CWS), and European paediatric Soft tissue sarcoma Study Group (EpSSG) to replace competing systems used in Europe and North America.( Hawkins, Bisogno, Koscielniak. Pediatr Blood Cancer. 2020;e28701.https://doi.org/10.1002/pbc.28701)

  1. Given the rate of secondary malignancies is identical to other series where G-CSF is used, the protocol mandate against growth factor should be discussed. Could the authors please address this? Were there any issues with treatment delay given the high rates of bone marrow toxicity documented in SR, HR, and VHR groups? Could this have affected outcome?

Response: SMN in patients with RMS and other STS can be associated with genetic predisposition, cumulative alkylators and anthracyclines doses, irradiation and also the use of G-CSF. (Caruso et al. PBC 2019) In our series, only 2 hematologic malignancies occurred (0.5%) that are thought to be associated with the prophylactic use of G-CSF (Lyman et al. Ann Oncol 2018) ). Bisogno et al. (Lancet Oncology 2018)  reported 2% ( IVA arm) and 4% ( IVA plus Doxo arm) SMN in the RMS2005 study but there is no information concerning the type of the SMN. The main reason for not using the G-CSF prophylactically was the lack of clinical need, based on our experience in the previous studies. A leucopenia Grade III-IV(1000-1900/µl) is not always life-threatening and is not always associated with severe neutropenia (<500/µl).  The rate of clinically documented infection was much lower (10-16%).  The use of G-CSF was recommended in clinically documented infection with neutropenia. We have not seen major treatment delays due to leucopenia that occurred mainly in the treatment interval. Since the topic concerning the indications for the prophylactic use of G-CSF, its possible impact on the outcome and the rate of second malignancies is very complex, we think that it would be would go beyond the scope of this analysis to discuss  all these aspects in this article.

  1. Lines 353-354 discuss aRMS with or without fusion status - please clarify this statement with the number (N) in each category. The low N makes this conclusion, which is contrary to established data, difficult to state conclusively and this should be made clear. The concluding line on line 453 also appears to make a conclusion out of this limited data, which should be adjusted or eliminated. It is unclear what "molecular-based risk parameters" the authors are suggesting would be useful and clarification is necessary. 

Response: In line with the reviewer's advice, we have clarified the information on aRMS with or without fusion status. As recommended, we have added the numbers of patients in each category: (probably lines 374-376 were meant).

“The distribution of histology in alveolar (n=74) vs. non-alveolar (n=355) compared to "fusion status positive (n=50) or negative (n=6) " did not yield a major difference in terms of prognostic significance since only six tumors with alveolar histology were "fusion negative".

We do not think that our data contradicts other published data. We only found that fusion status vs. pathology in the univariate analysis did not cause any fundamental change in EFS and OS (histology unfavorable n=74, 5-year EFS 59%, favorable n=355 5-year EFS 75%, Fusion positive n=50 5-year EFS 54% , Fusion negative (aRMS) n=6 5-year EFS 100%, Fusion unknown EFS 5-year 61% ) and the  fusion status was not a stronger predictor of poor outcome. The difference of our analysis to the published data like Hibbitts et al. Cancer Medicine 2019, is that by “fusion negative” we mean only the fusion negative alveolar RMS. Of course, the embryonal RMS are also probably fusion negative (although not all tested), but we wanted to examine the group of “fusion negative” that were diagnosed by our reference pathologists as clearly alveolar separately and not mix these 6 patients with the 355 embryonal RMS.

The diagnostic criteria for alveolar RMS used by our reference pathologists were different in comparisons to the MMT SIOP, EpSSG and COG groups and the rate of fusion negative aRMS was very low in our series. Since the pathology further plays an important diagnostic role, we think that it should be still taken into account.

Since only the most frequent fusion like Foxo1/Pax3/7 are routinely tested, it is still possible that other rare fusions like NCOA1,2/Pax3/7, PAX3-FOXO4,  FOXO1-FGFR1 might be present. Genomic analyses of RMS tumors show that other features, specifically TP53 m or MYOD1 mutations, may influence outcome. (Rudzinski et al. PBC 2020,Pediatr Blood Cancer) 2020;e28798. https://doi.org/10.1002/pbc.28798.   TP53 and MYOD1 Mutations present mostly in “fusion-negative” RMS but associated with very poor prognosis have been already included in the risk grouping by the COG. We therefore think that the reducing the RMS classification into "Foxo Fusion-positive vs. negative" has to be probably extended with other prognostic relevant markers, taking into account the deep heterogeneity of embryonal RMS. Missiaglia et al (JCO 2012) reported a very good prognosis of patients with Pax7/Foxo positive tumors, identical with fusion negative aRMS. There is therefore still an international effort needed to define, validate and introduce molecular markers into the classification of RMS. However, we agree with the reviewer that we cannot draw such a conclusion based on our present analysis. Since reviewer 1 recommended a comment concerning the anaplasia and spindle cell tumors, we have already added a statement on this.

We have adjusted the conclusion – as suggested by the reviewer.

“ In our opinion, the current risk stratification system in gemneral, which was historically developed through joint analyzes of potential risk factors within the framework of international cooperation should be reassessed. An international effort is needed before relevant biologic subtypes  may eventually be incorportated into standard practice. The first step to harmonize approaches to optimize RMS risk stratification is a collaboration of the three cooperative groups: Children's Oncology Group (COG), Cooperative Weichteilsarkom Studiengruppe (CWS), and European paediatric Soft tissue sarcoma Study Group (EpSSG) was initiated resulting in the INternational Soft Tissue SaRcoma ConsorTium (INSTRuCT) (https://commons.cri.uchicago.edu/instruct/)”

  1. The commentary on primary site grouping suggests making GU-BP favorable prognosis with discussion of available data to support this but appears to neglect discussing making HN-nPM unfavorable prognosis (Table S7). Could this latter point please be discussed further and if the literature supports this categorization. Furthermore, smaller p-values beyond the threshold of significance do not imply "more" significance and caution should be applied to interpreting this post-hoc evaluation of groupings in which multiple proposed groupings, including classical, are all significant.

Response: Our idea was to analyze the different grouping of primary sites, since the old grouping was based on the consensus reached many decades ago. However, we agree with the reviewer that p-values above the significance threshold do not imply "more significance" and we deleted Table S7. The new risk stratification must be based on more complex mathematical modeling of various parameters

Reviewer 2

This well-written and important paper summarizes the findings of iteratively-developed German study CWS-2002P, focusing on pediatric localized rhabdomyosarcoma patients on study from 2003 to 2010. The study was well-done with much valuable data generated for future study design and to establish the current European standard of care. The authors are to be commended for their efforts.

Should there be any question about the preponderance of localized PM disease in the H&N patients, the authors can refer readers to PMID: 10491544, which described older studies IRS-II and IRS-III, where 68% were cranial PM.

Minor questions arose while reviewing the paper:

  1. It seems that if confidence intervals for survival (EFS) with or without maintenance therapy (MT) are overlapping, it would be difficult if not impossible to achieve statistical significance. Could the authors clarify on this?

Response: If the confidence intervals partially overlap, it cannot be concluded that the parameters do not differ significantly. If the overlap is not too large (6%) and the number of patients not too small, the difference may still be significant. In our case resulting in a significant p of 0.015 (Source : Book Applied statistics L.Sachs u. J. Hedderich)

  1. Given significance of T-status in this study, will this be re-incorporated into future study risk stratification (Table S1)? Could the authors please discuss why T-status was removed from stratification?

Response: Facing this interesting point of view it has to be noticed that  T was a risk factor in the risk grouping of the CWS-96 study based on a European consensus between the study groups CWS, the Italian ICG STS Group and the MMT SIOP (and also in the ICG RMS-96 study and the MMT95 study has been used). A follow-up analysis performed on the CWS/RMS-96 data and validated on data from studies with longer follow-up: the SIOP-MMT-84 and -89 studies, the German CWS-81 - and -91 studies and the Italian RMS-79 and RMS-88 studies identified as significant: histology, IRS group, tumor location and size, N involvement, and age > 10 years, but not T status. The combination of these factors determined the risk grouping used in the CWS-2002P and EpSSG RMS2005. This grouping system is also used in the CWS guideline. The new EpSSG FaRMS trial (the CWS group will participate in the study) continues to use a similar risk grouping system without T status. In addition, the assignment of T1 and T2 by the local investigators was often incorrect (after a central assessment). It is therefore difficult to use it in a large international study. Nevertheless, we documented and centrally validated the T status in the CWS 2002P, and were able to incorporate it into the risk factor analysis. In summary, the decision to delete T status was a consensus decision, facilitating comparison between studies and reducing potential errors. A new, international risk grouping  for RMS system is  one of the goals of the INternational Soft TissueSaRcoma ConsorTium (INSTRUCT) composed of three large cooperative clinical trials organizations: Tissue

SaRcoma ConsorTium (INSTRuCT) (https://commons.cri.uchicago.edu/instruct/) Children's Oncology Group (COG), Cooperative Weichteilsarkom Studiengruppe (CWS), and European paediatric Soft tissue sarcoma Study Group (EpSSG) to replace competing systems used in Europe and North America.( Hawkins, Bisogno, Koscielniak. Pediatr Blood Cancer. 2020;e28701.https://doi.org/10.1002/pbc.28701)

  1. Given the rate of secondary malignancies is identical to other series where G-CSF is used, the protocol mandate against growth factor should be discussed. Could the authors please address this? Were there any issues with treatment delay given the high rates of bone marrow toxicity documented in SR, HR, and VHR groups? Could this have affected outcome?

Response: SMN in patients with RMS and other STS can be associated with genetic predisposition, cumulative alkylators and anthracyclines doses, irradiation and also the use of G-CSF. (Caruso et al. PBC 2019) In our series, only 2 hematologic malignancies occurred (0.5%) that are thought to be associated with the prophylactic use of G-CSF (Lyman et al. Ann Oncol 2018) ). Bisogno et al. (Lancet Oncology 2018)  reported 2% ( IVA arm) and 4% ( IVA plus Doxo arm) SMN in the RMS2005 study but there is no information concerning the type of the SMN. The main reason for not using the G-CSF prophylactically was the lack of clinical need, based on our experience in the previous studies. A leucopenia Grade III-IV(1000-1900/µl) is not always life-threatening and is not always associated with severe neutropenia (<500/µl).  The rate of clinically documented infection was much lower (10-16%).  The use of G-CSF was recommended in clinically documented infection with neutropenia. We have not seen major treatment delays due to leucopenia that occurred mainly in the treatment interval. Since the topic concerning the indications for the prophylactic use of G-CSF, its possible impact on the outcome and the rate of second malignancies is very complex, we think that it would be would go beyond the scope of this analysis to discuss  all these aspects in this article.

  1. Lines 353-354 discuss aRMS with or without fusion status - please clarify this statement with the number (N) in each category. The low N makes this conclusion, which is contrary to established data, difficult to state conclusively and this should be made clear. The concluding line on line 453 also appears to make a conclusion out of this limited data, which should be adjusted or eliminated. It is unclear what "molecular-based risk parameters" the authors are suggesting would be useful and clarification is necessary. 

Response: In line with the reviewer's advice, we have clarified the information on aRMS with or without fusion status. As recommended, we have added the numbers of patients in each category: (probably lines 374-376 were meant).

“The distribution of histology in alveolar (n=74) vs. non-alveolar (n=355) compared to "fusion status positive (n=50) or negative (n=6) " did not yield a major difference in terms of prognostic significance since only six tumors with alveolar histology were "fusion negative".

We do not think that our data contradicts other published data. We only found that fusion status vs. pathology in the univariate analysis did not cause any fundamental change in EFS and OS (histology unfavorable n=74, 5-year EFS 59%, favorable n=355 5-year EFS 75%, Fusion positive n=50 5-year EFS 54% , Fusion negative (aRMS) n=6 5-year EFS 100%, Fusion unknown EFS 5-year 61% ) and the  fusion status was not a stronger predictor of poor outcome. The difference of our analysis to the published data like Hibbitts et al. Cancer Medicine 2019, is that by “fusion negative” we mean only the fusion negative alveolar RMS. Of course, the embryonal RMS are also probably fusion negative (although not all tested), but we wanted to examine the group of “fusion negative” that were diagnosed by our reference pathologists as clearly alveolar separately and not mix these 6 patients with the 355 embryonal RMS.

The diagnostic criteria for alveolar RMS used by our reference pathologists were different in comparisons to the MMT SIOP, EpSSG and COG groups and the rate of fusion negative aRMS was very low in our series. Since the pathology further plays an important diagnostic role, we think that it should be still taken into account.

Since only the most frequent fusion like Foxo1/Pax3/7 are routinely tested, it is still possible that other rare fusions like NCOA1,2/Pax3/7, PAX3-FOXO4,  FOXO1-FGFR1 might be present. Genomic analyses of RMS tumors show that other features, specifically TP53 m or MYOD1 mutations, may influence outcome. (Rudzinski et al. PBC 2020,Pediatr Blood Cancer) 2020;e28798. https://doi.org/10.1002/pbc.28798.   TP53 and MYOD1 Mutations present mostly in “fusion-negative” RMS but associated with very poor prognosis have been already included in the risk grouping by the COG. We therefore think that the reducing the RMS classification into "Foxo Fusion-positive vs. negative" has to be probably extended with other prognostic relevant markers, taking into account the deep heterogeneity of embryonal RMS. Missiaglia et al (JCO 2012) reported a very good prognosis of patients with Pax7/Foxo positive tumors, identical with fusion negative aRMS. There is therefore still an international effort needed to define, validate and introduce molecular markers into the classification of RMS. However, we agree with the reviewer that we cannot draw such a conclusion based on our present analysis. Since reviewer 1 recommended a comment concerning the anaplasia and spindle cell tumors, we have already added a statement on this.

We have adjusted the conclusion – as suggested by the reviewer.

“ In our opinion, the current risk stratification system in gemneral, which was historically developed through joint analyzes of potential risk factors within the framework of international cooperation should be reassessed. An international effort is needed before relevant biologic subtypes  may eventually be incorportated into standard practice. The first step to harmonize approaches to optimize RMS risk stratification is a collaboration of the three cooperative groups: Children's Oncology Group (COG), Cooperative Weichteilsarkom Studiengruppe (CWS), and European paediatric Soft tissue sarcoma Study Group (EpSSG) was initiated resulting in the INternational Soft Tissue SaRcoma ConsorTium (INSTRuCT) (https://commons.cri.uchicago.edu/instruct/)”

  1. The commentary on primary site grouping suggests making GU-BP favorable prognosis with discussion of available data to support this but appears to neglect discussing making HN-nPM unfavorable prognosis (Table S7). Could this latter point please be discussed further and if the literature supports this categorization. Furthermore, smaller p-values beyond the threshold of significance do not imply "more" significance and caution should be applied to interpreting this post-hoc evaluation of groupings in which multiple proposed groupings, including classical, are all significant.

Response: Our idea was to analyze the different grouping of primary sites, since the old grouping was based on the consensus reached many decades ago. However, we agree with the reviewer that p-values above the significance threshold do not imply "more significance" and we deleted Table S7. The new risk stratification must be based on more complex mathematical modeling of various parameters

Reviewer 2

This well-written and important paper summarizes the findings of iteratively-developed German study CWS-2002P, focusing on pediatric localized rhabdomyosarcoma patients on study from 2003 to 2010. The study was well-done with much valuable data generated for future study design and to establish the current European standard of care. The authors are to be commended for their efforts.

Should there be any question about the preponderance of localized PM disease in the H&N patients, the authors can refer readers to PMID: 10491544, which described older studies IRS-II and IRS-III, where 68% were cranial PM.

Minor questions arose while reviewing the paper:

  1. It seems that if confidence intervals for survival (EFS) with or without maintenance therapy (MT) are overlapping, it would be difficult if not impossible to achieve statistical significance. Could the authors clarify on this?

Response: If the confidence intervals partially overlap, it cannot be concluded that the parameters do not differ significantly. If the overlap is not too large (6%) and the number of patients not too small, the difference may still be significant. In our case resulting in a significant p of 0.015 (Source : Book Applied statistics L.Sachs u. J. Hedderich)

  1. Given significance of T-status in this study, will this be re-incorporated into future study risk stratification (Table S1)? Could the authors please discuss why T-status was removed from stratification?

Response: Facing this interesting point of view it has to be noticed that  T was a risk factor in the risk grouping of the CWS-96 study based on a European consensus between the study groups CWS, the Italian ICG STS Group and the MMT SIOP (and also in the ICG RMS-96 study and the MMT95 study has been used). A follow-up analysis performed on the CWS/RMS-96 data and validated on data from studies with longer follow-up: the SIOP-MMT-84 and -89 studies, the German CWS-81 - and -91 studies and the Italian RMS-79 and RMS-88 studies identified as significant: histology, IRS group, tumor location and size, N involvement, and age > 10 years, but not T status. The combination of these factors determined the risk grouping used in the CWS-2002P and EpSSG RMS2005. This grouping system is also used in the CWS guideline. The new EpSSG FaRMS trial (the CWS group will participate in the study) continues to use a similar risk grouping system without T status. In addition, the assignment of T1 and T2 by the local investigators was often incorrect (after a central assessment). It is therefore difficult to use it in a large international study. Nevertheless, we documented and centrally validated the T status in the CWS 2002P, and were able to incorporate it into the risk factor analysis. In summary, the decision to delete T status was a consensus decision, facilitating comparison between studies and reducing potential errors. A new, international risk grouping  for RMS system is  one of the goals of the INternational Soft TissueSaRcoma ConsorTium (INSTRUCT) composed of three large cooperative clinical trials organizations: Tissue

SaRcoma ConsorTium (INSTRuCT) (https://commons.cri.uchicago.edu/instruct/) Children's Oncology Group (COG), Cooperative Weichteilsarkom Studiengruppe (CWS), and European paediatric Soft tissue sarcoma Study Group (EpSSG) to replace competing systems used in Europe and North America.( Hawkins, Bisogno, Koscielniak. Pediatr Blood Cancer. 2020;e28701.https://doi.org/10.1002/pbc.28701)

  1. Given the rate of secondary malignancies is identical to other series where G-CSF is used, the protocol mandate against growth factor should be discussed. Could the authors please address this? Were there any issues with treatment delay given the high rates of bone marrow toxicity documented in SR, HR, and VHR groups? Could this have affected outcome?

Response: SMN in patients with RMS and other STS can be associated with genetic predisposition, cumulative alkylators and anthracyclines doses, irradiation and also the use of G-CSF. (Caruso et al. PBC 2019) In our series, only 2 hematologic malignancies occurred (0.5%) that are thought to be associated with the prophylactic use of G-CSF (Lyman et al. Ann Oncol 2018) ). Bisogno et al. (Lancet Oncology 2018)  reported 2% ( IVA arm) and 4% ( IVA plus Doxo arm) SMN in the RMS2005 study but there is no information concerning the type of the SMN. The main reason for not using the G-CSF prophylactically was the lack of clinical need, based on our experience in the previous studies. A leucopenia Grade III-IV(1000-1900/µl) is not always life-threatening and is not always associated with severe neutropenia (<500/µl).  The rate of clinically documented infection was much lower (10-16%).  The use of G-CSF was recommended in clinically documented infection with neutropenia. We have not seen major treatment delays due to leucopenia that occurred mainly in the treatment interval. Since the topic concerning the indications for the prophylactic use of G-CSF, its possible impact on the outcome and the rate of second malignancies is very complex, we think that it would be would go beyond the scope of this analysis to discuss  all these aspects in this article.

  1. Lines 353-354 discuss aRMS with or without fusion status - please clarify this statement with the number (N) in each category. The low N makes this conclusion, which is contrary to established data, difficult to state conclusively and this should be made clear. The concluding line on line 453 also appears to make a conclusion out of this limited data, which should be adjusted or eliminated. It is unclear what "molecular-based risk parameters" the authors are suggesting would be useful and clarification is necessary. 

Response: In line with the reviewer's advice, we have clarified the information on aRMS with or without fusion status. As recommended, we have added the numbers of patients in each category: (probably lines 374-376 were meant).

“The distribution of histology in alveolar (n=74) vs. non-alveolar (n=355) compared to "fusion status positive (n=50) or negative (n=6) " did not yield a major difference in terms of prognostic significance since only six tumors with alveolar histology were "fusion negative".

We do not think that our data contradicts other published data. We only found that fusion status vs. pathology in the univariate analysis did not cause any fundamental change in EFS and OS (histology unfavorable n=74, 5-year EFS 59%, favorable n=355 5-year EFS 75%, Fusion positive n=50 5-year EFS 54% , Fusion negative (aRMS) n=6 5-year EFS 100%, Fusion unknown EFS 5-year 61% ) and the  fusion status was not a stronger predictor of poor outcome. The difference of our analysis to the published data like Hibbitts et al. Cancer Medicine 2019, is that by “fusion negative” we mean only the fusion negative alveolar RMS. Of course, the embryonal RMS are also probably fusion negative (although not all tested), but we wanted to examine the group of “fusion negative” that were diagnosed by our reference pathologists as clearly alveolar separately and not mix these 6 patients with the 355 embryonal RMS.

The diagnostic criteria for alveolar RMS used by our reference pathologists were different in comparisons to the MMT SIOP, EpSSG and COG groups and the rate of fusion negative aRMS was very low in our series. Since the pathology further plays an important diagnostic role, we think that it should be still taken into account.

Since only the most frequent fusion like Foxo1/Pax3/7 are routinely tested, it is still possible that other rare fusions like NCOA1,2/Pax3/7, PAX3-FOXO4,  FOXO1-FGFR1 might be present. Genomic analyses of RMS tumors show that other features, specifically TP53 m or MYOD1 mutations, may influence outcome. (Rudzinski et al. PBC 2020,Pediatr Blood Cancer) 2020;e28798. https://doi.org/10.1002/pbc.28798.   TP53 and MYOD1 Mutations present mostly in “fusion-negative” RMS but associated with very poor prognosis have been already included in the risk grouping by the COG. We therefore think that the reducing the RMS classification into "Foxo Fusion-positive vs. negative" has to be probably extended with other prognostic relevant markers, taking into account the deep heterogeneity of embryonal RMS. Missiaglia et al (JCO 2012) reported a very good prognosis of patients with Pax7/Foxo positive tumors, identical with fusion negative aRMS. There is therefore still an international effort needed to define, validate and introduce molecular markers into the classification of RMS. However, we agree with the reviewer that we cannot draw such a conclusion based on our present analysis. Since reviewer 1 recommended a comment concerning the anaplasia and spindle cell tumors, we have already added a statement on this.

We have adjusted the conclusion – as suggested by the reviewer.

“ In our opinion, the current risk stratification system in gemneral, which was historically developed through joint analyzes of potential risk factors within the framework of international cooperation should be reassessed. An international effort is needed before relevant biologic subtypes  may eventually be incorportated into standard practice. The first step to harmonize approaches to optimize RMS risk stratification is a collaboration of the three cooperative groups: Children's Oncology Group (COG), Cooperative Weichteilsarkom Studiengruppe (CWS), and European paediatric Soft tissue sarcoma Study Group (EpSSG) was initiated resulting in the INternational Soft Tissue SaRcoma ConsorTium (INSTRuCT) (https://commons.cri.uchicago.edu/instruct/)”

  1. The commentary on primary site grouping suggests making GU-BP favorable prognosis with discussion of available data to support this but appears to neglect discussing making HN-nPM unfavorable prognosis (Table S7). Could this latter point please be discussed further and if the literature supports this categorization. Furthermore, smaller p-values beyond the threshold of significance do not imply "more" significance and caution should be applied to interpreting this post-hoc evaluation of groupings in which multiple proposed groupings, including classical, are all significant.

Response: Our idea was to analyze the different grouping of primary sites, since the old grouping was based on the consensus reached many decades ago. However, we agree with the reviewer that p-values above the significance threshold do not imply "more significance" and we deleted Table S7. The new risk stratification must be based on more complex mathematical modeling of various parameters

Reviewer 2

This well-written and important paper summarizes the findings of iteratively-developed German study CWS-2002P, focusing on pediatric localized rhabdomyosarcoma patients on study from 2003 to 2010. The study was well-done with much valuable data generated for future study design and to establish the current European standard of care. The authors are to be commended for their efforts.

Should there be any question about the preponderance of localized PM disease in the H&N patients, the authors can refer readers to PMID: 10491544, which described older studies IRS-II and IRS-III, where 68% were cranial PM.

Minor questions arose while reviewing the paper:

  1. It seems that if confidence intervals for survival (EFS) with or without maintenance therapy (MT) are overlapping, it would be difficult if not impossible to achieve statistical significance. Could the authors clarify on this?

Response: If the confidence intervals partially overlap, it cannot be concluded that the parameters do not differ significantly. If the overlap is not too large (6%) and the number of patients not too small, the difference may still be significant. In our case resulting in a significant p of 0.015 (Source : Book Applied statistics L.Sachs u. J. Hedderich)

  1. Given significance of T-status in this study, will this be re-incorporated into future study risk stratification (Table S1)? Could the authors please discuss why T-status was removed from stratification?

Response: Facing this interesting point of view it has to be noticed that  T was a risk factor in the risk grouping of the CWS-96 study based on a European consensus between the study groups CWS, the Italian ICG STS Group and the MMT SIOP (and also in the ICG RMS-96 study and the MMT95 study has been used). A follow-up analysis performed on the CWS/RMS-96 data and validated on data from studies with longer follow-up: the SIOP-MMT-84 and -89 studies, the German CWS-81 - and -91 studies and the Italian RMS-79 and RMS-88 studies identified as significant: histology, IRS group, tumor location and size, N involvement, and age > 10 years, but not T status. The combination of these factors determined the risk grouping used in the CWS-2002P and EpSSG RMS2005. This grouping system is also used in the CWS guideline. The new EpSSG FaRMS trial (the CWS group will participate in the study) continues to use a similar risk grouping system without T status. In addition, the assignment of T1 and T2 by the local investigators was often incorrect (after a central assessment). It is therefore difficult to use it in a large international study. Nevertheless, we documented and centrally validated the T status in the CWS 2002P, and were able to incorporate it into the risk factor analysis. In summary, the decision to delete T status was a consensus decision, facilitating comparison between studies and reducing potential errors. A new, international risk grouping  for RMS system is  one of the goals of the INternational Soft TissueSaRcoma ConsorTium (INSTRUCT) composed of three large cooperative clinical trials organizations: Tissue

SaRcoma ConsorTium (INSTRuCT) (https://commons.cri.uchicago.edu/instruct/) Children's Oncology Group (COG), Cooperative Weichteilsarkom Studiengruppe (CWS), and European paediatric Soft tissue sarcoma Study Group (EpSSG) to replace competing systems used in Europe and North America.( Hawkins, Bisogno, Koscielniak. Pediatr Blood Cancer. 2020;e28701.https://doi.org/10.1002/pbc.28701)

  1. Given the rate of secondary malignancies is identical to other series where G-CSF is used, the protocol mandate against growth factor should be discussed. Could the authors please address this? Were there any issues with treatment delay given the high rates of bone marrow toxicity documented in SR, HR, and VHR groups? Could this have affected outcome?

Response: SMN in patients with RMS and other STS can be associated with genetic predisposition, cumulative alkylators and anthracyclines doses, irradiation and also the use of G-CSF. (Caruso et al. PBC 2019) In our series, only 2 hematologic malignancies occurred (0.5%) that are thought to be associated with the prophylactic use of G-CSF (Lyman et al. Ann Oncol 2018) ). Bisogno et al. (Lancet Oncology 2018)  reported 2% ( IVA arm) and 4% ( IVA plus Doxo arm) SMN in the RMS2005 study but there is no information concerning the type of the SMN. The main reason for not using the G-CSF prophylactically was the lack of clinical need, based on our experience in the previous studies. A leucopenia Grade III-IV(1000-1900/µl) is not always life-threatening and is not always associated with severe neutropenia (<500/µl).  The rate of clinically documented infection was much lower (10-16%).  The use of G-CSF was recommended in clinically documented infection with neutropenia. We have not seen major treatment delays due to leucopenia that occurred mainly in the treatment interval. Since the topic concerning the indications for the prophylactic use of G-CSF, its possible impact on the outcome and the rate of second malignancies is very complex, we think that it would be would go beyond the scope of this analysis to discuss  all these aspects in this article.

  1. Lines 353-354 discuss aRMS with or without fusion status - please clarify this statement with the number (N) in each category. The low N makes this conclusion, which is contrary to established data, difficult to state conclusively and this should be made clear. The concluding line on line 453 also appears to make a conclusion out of this limited data, which should be adjusted or eliminated. It is unclear what "molecular-based risk parameters" the authors are suggesting would be useful and clarification is necessary. 

Response: In line with the reviewer's advice, we have clarified the information on aRMS with or without fusion status. As recommended, we have added the numbers of patients in each category: (probably lines 374-376 were meant).

“The distribution of histology in alveolar (n=74) vs. non-alveolar (n=355) compared to "fusion status positive (n=50) or negative (n=6) " did not yield a major difference in terms of prognostic significance since only six tumors with alveolar histology were "fusion negative".

We do not think that our data contradicts other published data. We only found that fusion status vs. pathology in the univariate analysis did not cause any fundamental change in EFS and OS (histology unfavorable n=74, 5-year EFS 59%, favorable n=355 5-year EFS 75%, Fusion positive n=50 5-year EFS 54% , Fusion negative (aRMS) n=6 5-year EFS 100%, Fusion unknown EFS 5-year 61% ) and the  fusion status was not a stronger predictor of poor outcome. The difference of our analysis to the published data like Hibbitts et al. Cancer Medicine 2019, is that by “fusion negative” we mean only the fusion negative alveolar RMS. Of course, the embryonal RMS are also probably fusion negative (although not all tested), but we wanted to examine the group of “fusion negative” that were diagnosed by our reference pathologists as clearly alveolar separately and not mix these 6 patients with the 355 embryonal RMS.

The diagnostic criteria for alveolar RMS used by our reference pathologists were different in comparisons to the MMT SIOP, EpSSG and COG groups and the rate of fusion negative aRMS was very low in our series. Since the pathology further plays an important diagnostic role, we think that it should be still taken into account.

Since only the most frequent fusion like Foxo1/Pax3/7 are routinely tested, it is still possible that other rare fusions like NCOA1,2/Pax3/7, PAX3-FOXO4,  FOXO1-FGFR1 might be present. Genomic analyses of RMS tumors show that other features, specifically TP53 m or MYOD1 mutations, may influence outcome. (Rudzinski et al. PBC 2020,Pediatr Blood Cancer) 2020;e28798. https://doi.org/10.1002/pbc.28798.   TP53 and MYOD1 Mutations present mostly in “fusion-negative” RMS but associated with very poor prognosis have been already included in the risk grouping by the COG. We therefore think that the reducing the RMS classification into "Foxo Fusion-positive vs. negative" has to be probably extended with other prognostic relevant markers, taking into account the deep heterogeneity of embryonal RMS. Missiaglia et al (JCO 2012) reported a very good prognosis of patients with Pax7/Foxo positive tumors, identical with fusion negative aRMS. There is therefore still an international effort needed to define, validate and introduce molecular markers into the classification of RMS. However, we agree with the reviewer that we cannot draw such a conclusion based on our present analysis. Since reviewer 1 recommended a comment concerning the anaplasia and spindle cell tumors, we have already added a statement on this.

We have adjusted the conclusion – as suggested by the reviewer.

“ In our opinion, the current risk stratification system in gemneral, which was historically developed through joint analyzes of potential risk factors within the framework of international cooperation should be reassessed. An international effort is needed before relevant biologic subtypes  may eventually be incorportated into standard practice. The first step to harmonize approaches to optimize RMS risk stratification is a collaboration of the three cooperative groups: Children's Oncology Group (COG), Cooperative Weichteilsarkom Studiengruppe (CWS), and European paediatric Soft tissue sarcoma Study Group (EpSSG) was initiated resulting in the INternational Soft Tissue SaRcoma ConsorTium (INSTRuCT) (https://commons.cri.uchicago.edu/instruct/)”

  1. The commentary on primary site grouping suggests making GU-BP favorable prognosis with discussion of available data to support this but appears to neglect discussing making HN-nPM unfavorable prognosis (Table S7). Could this latter point please be discussed further and if the literature supports this categorization. Furthermore, smaller p-values beyond the threshold of significance do not imply "more" significance and caution should be applied to interpreting this post-hoc evaluation of groupings in which multiple proposed groupings, including classical, are all significant.

Response: Our idea was to analyze the different grouping of primary sites, since the old grouping was based on the consensus reached many decades ago. However, we agree with the reviewer that p-values above the significance threshold do not imply "more significance" and we deleted Table S7. The new risk stratification must be based on more complex mathematical modeling of various parameters

Reviewer 2

This well-written and important paper summarizes the findings of iteratively-developed German study CWS-2002P, focusing on pediatric localized rhabdomyosarcoma patients on study from 2003 to 2010. The study was well-done with much valuable data generated for future study design and to establish the current European standard of care. The authors are to be commended for their efforts.

Should there be any question about the preponderance of localized PM disease in the H&N patients, the authors can refer readers to PMID: 10491544, which described older studies IRS-II and IRS-III, where 68% were cranial PM.

Minor questions arose while reviewing the paper:

  1. It seems that if confidence intervals for survival (EFS) with or without maintenance therapy (MT) are overlapping, it would be difficult if not impossible to achieve statistical significance. Could the authors clarify on this?

Response: If the confidence intervals partially overlap, it cannot be concluded that the parameters do not differ significantly. If the overlap is not too large (6%) and the number of patients not too small, the difference may still be significant. In our case resulting in a significant p of 0.015 (Source : Book Applied statistics L.Sachs u. J. Hedderich)

  1. Given significance of T-status in this study, will this be re-incorporated into future study risk stratification (Table S1)? Could the authors please discuss why T-status was removed from stratification?

Response: Facing this interesting point of view it has to be noticed that  T was a risk factor in the risk grouping of the CWS-96 study based on a European consensus between the study groups CWS, the Italian ICG STS Group and the MMT SIOP (and also in the ICG RMS-96 study and the MMT95 study has been used). A follow-up analysis performed on the CWS/RMS-96 data and validated on data from studies with longer follow-up: the SIOP-MMT-84 and -89 studies, the German CWS-81 - and -91 studies and the Italian RMS-79 and RMS-88 studies identified as significant: histology, IRS group, tumor location and size, N involvement, and age > 10 years, but not T status. The combination of these factors determined the risk grouping used in the CWS-2002P and EpSSG RMS2005. This grouping system is also used in the CWS guideline. The new EpSSG FaRMS trial (the CWS group will participate in the study) continues to use a similar risk grouping system without T status. In addition, the assignment of T1 and T2 by the local investigators was often incorrect (after a central assessment). It is therefore difficult to use it in a large international study. Nevertheless, we documented and centrally validated the T status in the CWS 2002P, and were able to incorporate it into the risk factor analysis. In summary, the decision to delete T status was a consensus decision, facilitating comparison between studies and reducing potential errors. A new, international risk grouping  for RMS system is  one of the goals of the INternational Soft TissueSaRcoma ConsorTium (INSTRUCT) composed of three large cooperative clinical trials organizations: Tissue

SaRcoma ConsorTium (INSTRuCT) (https://commons.cri.uchicago.edu/instruct/) Children's Oncology Group (COG), Cooperative Weichteilsarkom Studiengruppe (CWS), and European paediatric Soft tissue sarcoma Study Group (EpSSG) to replace competing systems used in Europe and North America.( Hawkins, Bisogno, Koscielniak. Pediatr Blood Cancer. 2020;e28701.https://doi.org/10.1002/pbc.28701)

  1. Given the rate of secondary malignancies is identical to other series where G-CSF is used, the protocol mandate against growth factor should be discussed. Could the authors please address this? Were there any issues with treatment delay given the high rates of bone marrow toxicity documented in SR, HR, and VHR groups? Could this have affected outcome?

Response: SMN in patients with RMS and other STS can be associated with genetic predisposition, cumulative alkylators and anthracyclines doses, irradiation and also the use of G-CSF. (Caruso et al. PBC 2019) In our series, only 2 hematologic malignancies occurred (0.5%) that are thought to be associated with the prophylactic use of G-CSF (Lyman et al. Ann Oncol 2018) ). Bisogno et al. (Lancet Oncology 2018)  reported 2% ( IVA arm) and 4% ( IVA plus Doxo arm) SMN in the RMS2005 study but there is no information concerning the type of the SMN. The main reason for not using the G-CSF prophylactically was the lack of clinical need, based on our experience in the previous studies. A leucopenia Grade III-IV(1000-1900/µl) is not always life-threatening and is not always associated with severe neutropenia (<500/µl).  The rate of clinically documented infection was much lower (10-16%).  The use of G-CSF was recommended in clinically documented infection with neutropenia. We have not seen major treatment delays due to leucopenia that occurred mainly in the treatment interval. Since the topic concerning the indications for the prophylactic use of G-CSF, its possible impact on the outcome and the rate of second malignancies is very complex, we think that it would be would go beyond the scope of this analysis to discuss  all these aspects in this article.

  1. Lines 353-354 discuss aRMS with or without fusion status - please clarify this statement with the number (N) in each category. The low N makes this conclusion, which is contrary to established data, difficult to state conclusively and this should be made clear. The concluding line on line 453 also appears to make a conclusion out of this limited data, which should be adjusted or eliminated. It is unclear what "molecular-based risk parameters" the authors are suggesting would be useful and clarification is necessary. 

Response: In line with the reviewer's advice, we have clarified the information on aRMS with or without fusion status. As recommended, we have added the numbers of patients in each category: (probably lines 374-376 were meant).

“The distribution of histology in alveolar (n=74) vs. non-alveolar (n=355) compared to "fusion status positive (n=50) or negative (n=6) " did not yield a major difference in terms of prognostic significance since only six tumors with alveolar histology were "fusion negative".

We do not think that our data contradicts other published data. We only found that fusion status vs. pathology in the univariate analysis did not cause any fundamental change in EFS and OS (histology unfavorable n=74, 5-year EFS 59%, favorable n=355 5-year EFS 75%, Fusion positive n=50 5-year EFS 54% , Fusion negative (aRMS) n=6 5-year EFS 100%, Fusion unknown EFS 5-year 61% ) and the  fusion status was not a stronger predictor of poor outcome. The difference of our analysis to the published data like Hibbitts et al. Cancer Medicine 2019, is that by “fusion negative” we mean only the fusion negative alveolar RMS. Of course, the embryonal RMS are also probably fusion negative (although not all tested), but we wanted to examine the group of “fusion negative” that were diagnosed by our reference pathologists as clearly alveolar separately and not mix these 6 patients with the 355 embryonal RMS.

The diagnostic criteria for alveolar RMS used by our reference pathologists were different in comparisons to the MMT SIOP, EpSSG and COG groups and the rate of fusion negative aRMS was very low in our series. Since the pathology further plays an important diagnostic role, we think that it should be still taken into account.

Since only the most frequent fusion like Foxo1/Pax3/7 are routinely tested, it is still possible that other rare fusions like NCOA1,2/Pax3/7, PAX3-FOXO4,  FOXO1-FGFR1 might be present. Genomic analyses of RMS tumors show that other features, specifically TP53 m or MYOD1 mutations, may influence outcome. (Rudzinski et al. PBC 2020,Pediatr Blood Cancer) 2020;e28798. https://doi.org/10.1002/pbc.28798.   TP53 and MYOD1 Mutations present mostly in “fusion-negative” RMS but associated with very poor prognosis have been already included in the risk grouping by the COG. We therefore think that the reducing the RMS classification into "Foxo Fusion-positive vs. negative" has to be probably extended with other prognostic relevant markers, taking into account the deep heterogeneity of embryonal RMS. Missiaglia et al (JCO 2012) reported a very good prognosis of patients with Pax7/Foxo positive tumors, identical with fusion negative aRMS. There is therefore still an international effort needed to define, validate and introduce molecular markers into the classification of RMS. However, we agree with the reviewer that we cannot draw such a conclusion based on our present analysis. Since reviewer 1 recommended a comment concerning the anaplasia and spindle cell tumors, we have already added a statement on this.

We have adjusted the conclusion – as suggested by the reviewer.

“ In our opinion, the current risk stratification system in gemneral, which was historically developed through joint analyzes of potential risk factors within the framework of international cooperation should be reassessed. An international effort is needed before relevant biologic subtypes  may eventually be incorportated into standard practice. The first step to harmonize approaches to optimize RMS risk stratification is a collaboration of the three cooperative groups: Children's Oncology Group (COG), Cooperative Weichteilsarkom Studiengruppe (CWS), and European paediatric Soft tissue sarcoma Study Group (EpSSG) was initiated resulting in the INternational Soft Tissue SaRcoma ConsorTium (INSTRuCT) (https://commons.cri.uchicago.edu/instruct/)”

  1. The commentary on primary site grouping suggests making GU-BP favorable prognosis with discussion of available data to support this but appears to neglect discussing making HN-nPM unfavorable prognosis (Table S7). Could this latter point please be discussed further and if the literature supports this categorization. Furthermore, smaller p-values beyond the threshold of significance do not imply "more" significance and caution should be applied to interpreting this post-hoc evaluation of groupings in which multiple proposed groupings, including classical, are all significant.

Response: Our idea was to analyze the different grouping of primary sites, since the old grouping was based on the consensus reached many decades ago. However, we agree with the reviewer that p-values above the significance threshold do not imply "more significance" and we deleted Table S7. The new risk stratification must be based on more complex mathematical modeling of various parameters